# Surface ocean pH variations since 1689 CE and recent ocean acidification in the tropical South Pacific

Henry C. Wu [1,2,3], Delphine Dissard [1], Eric Douville [2], Dominique Blamart [2], Louise Bordier[2], Aline Tribollet[1], Florence Le Cornec [1], Edwige Pons-Branchu [2], Arnaud Dapoigny[2] & Claire E. Lazareth [1]

Increasing atmospheric $CO_2$ from man-made climate change is reducing surface ocean pH. Due to limited instrumental measurements and historical pH records in the world's oceans, seawater pH variability at the decadal and centennial scale remains largely unknown and requires documentation. Here we present evidence of striking secular trends of decreasing pH since the late nineteenth century with pronounced interannual to decadal–interdecadal pH variability in the South Pacific Ocean from 1689 to 2011 CE. High-amplitude oceanic pH changes, likely related to atmospheric $CO_2$ uptake and seawater dissolved inorganic carbon fluctuations, reveal a coupled relationship to sea surface temperature variations and highlight the marked influence of El Niño/Southern Oscillation and Interdecadal Pacific Oscillation. We suggest changing surface winds strength and zonal advection processes as the main drivers responsible for regional pH variability up to 1881 CE, followed by the prominent role of anthropogenic $CO_2$ in accelerating the process of ocean acidification.

[1] Institut de Recherche pour le Développement (IRD), Sorbonne Universités (UPMC Université Paris 06, CNRS, MNHN), UMR LOCEAN/IPSL, IRD DR Ile-de-France, 32 Avenue Henri Varagnat, F-93143 Bondy, France. [2] Laboratoire des Sciences du Climat et de l'Environnement, LSCE/IPSL, CEA-CNRS-UVSQ, Université Paris-Saclay, Bât. 12, Avenue de la Terrasse, F-91198 Gif-sur-Yvette, France. [3] Present address: Leibniz Centre for Tropical Marine Research (ZMT) GmbH, Fahrenheitstraße 6, D-28359 Bremen, Germany. Correspondence and requests for materials should be addressed to H.C.W. (email: henry.wu@leibniz-zmt.de)

The concentration of atmospheric $CO_2$ is increasing at unrelenting rates in response to human activities in fossil fuel combustion and land-use practices[1]. At present, the oceans take up more than 41% of anthropogenic $CO_2$ emissions[2], inducing the acidification of the ocean surface with major implications in marine carbonate chemistry and biological ecosystems[3]. Many modelling and experimental studies have indeed shown clear trends of shallow water acidification in lockstep with increasing atmospheric $CO_2$[4]. One of the major consequences is the decrease of carbonate saturation state that is crucial for calcifying organisms such as scleractinian corals to precipitate their aragonite skeletons[5]. These critical threats of ocean acidification (OA) on such marine organisms and ecosystems have been documented on the Great Barrier Reef (GBR) in Australia that witnessed a 14% decrease in coral calcification since 1990[6]. Future projections of coral calcification rates suggest a possible additional decrease by up to 30% in the twenty-first century[7]. Moreover, end-of-century emission scenarios predicted by model simulations indicate unprecedented decrease by up to 0.4 pH units[8] compared to the changes of only 0.2 pH units over glacial-interglacial cycles[9].

Presently, accurate measurements of pH in the oceans are near non-existent aside from selected ship track measurements[10] and limited station-based instrumental time series available over the past few decades[11]. The pH records that span only the most recent decades are far too short to precisely monitor the longer-term evolution of OA and to quantify its impacts on tropical ecosystems. To study the natural variability in oceanic $CO_2$ uptake and carbonate chemistry changes, proxy records in modern marine archives such as corals provide a unique opportunity to go further back in time at a high resolution on the centennial scale. Additionally, it is crucial to investigate pH changes with links to ocean $CO_2$ uptake as well as the relationship to sea surface temperature and forcing mechanisms at various timescales.

A method to document sub-seasonal to annual pH changes is based on the $\delta^{11}B$ signature in scleractinian coral skeleton. Coral-based reconstructions of ocean pH suggest notable recent acidification[12,13] and decadal variability[12,14]. However, due to spatiotemporal constrains, the currently available coral-based pH reconstructions predominantly focus on the GBR and the marginal South China Sea[12–16], whereas longer timescale pH changes of the open surface ocean from the Pacific Ocean remain poorly documented[17].

In the south-western Pacific, large-scale atmospheric circulation features and the seasonally varying latitudinal position of the tropical convergence zones (Intertropical and South Pacific Convergence Zones (ITCZ and SPCZ) extending from the western Pacific warm pool (WPWP; Fig. 1) are primarily controlling climate variability. The large-scale atmospheric circulation features include the seasonal trade wind regime and strength in addition to the Hadley and Walker Circulations. The region is also intimately pulsed at the interannual timescale by the Pacific basin-wide phenomena of El Niño/Southern Oscillation (ENSO) and at decadal timescale by the Interdecadal Pacific Oscillation (IPO) both in relation to the modulation of Pacific sea surface temperature (SST). More specifically, New Caledonia and the surrounding region of the south-western Pacific are of great interest for investigation because more than 40% of the global anthropogenic $CO_2$ inventory is found in the region between 14°S and 50°S[18] and about 60% of the total oceanic anthropogenic $CO_2$ is incorporated in the Southern Hemisphere oceans[4] (Fig. 1). Moreover, prior geochemical studies with annually banded scleractinian corals have demonstrated its advantages as reliable recorders of climate change from this region[19].

Here we present 323 years of annually resolved $\delta^{18}O$, $\delta^{13}C$ and $\delta^{11}B$ results from a modern slow-growing massive scleractinian coral, *Diploastrea heliopora*, collected from the open-ocean island of New Caledonia in the South Pacific Ocean (22.21°S, 166.15°E; Fig. 1 and Supplementary Fig. 1). In this study, we provide evidence of the uninterrupted OA with secular decreasing pH trend in the South Pacific since the Industrial Revolution. We show that pronounced interannual to decadal-interdecadal pH variability has been regularly occurring in the South Pacific since 1689 CE. The high-amplitude oceanic pH changes over these timescales are likely related to atmospheric $CO_2$ uptake because of significant coherence to coral $\delta^{13}C$ signature that tracked the changes in oceanic dissolved inorganic carbon (DIC) fluctuations. In addition, the timing of reconstructed pH variability in the South Pacific is coupled to prominent sea surface temperature

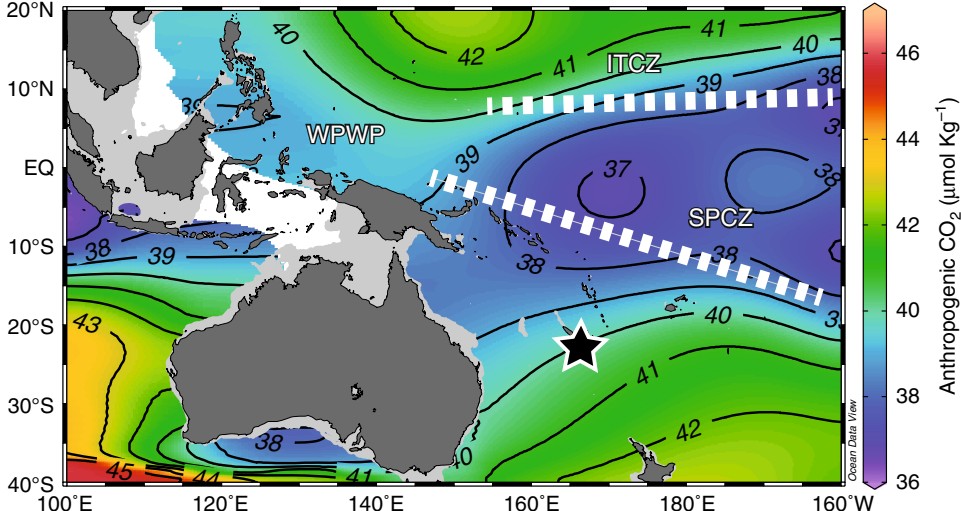

**Fig. 1** Research location. Our research location at New Caledonia (22°21′47 S, 166°15′29 E; Star; Supplementary Fig. 1) is of high importance because it is located in the zone of transition from low to high amounts of anthropogenic $CO_2$ uptake[73]. The western Pacific warm pool (WPWP) with a permanent sea surface temperature above 28 °C influences the climate of New Caledonia driving the steady convection in this region corresponding to the average precipitation axes of the Intertropical Convergence Zone (ITCZ) and South Pacific Convergence Zone (SPCZ). Map is produced using Ocean Data View ver. 4.7.4 (ref. [74]) (http://odv.awi.de) and modified manually

variations modulated by the phases of ENSO and IPO. We suggest changing surface wind strengths and zonal advection processes across the Pacific Ocean as the main drivers responsible for South Pacific pH variability that is superimposed by the prominent role of anthropogenic $CO_2$ increase in accelerating the process of OA.

## Results

**New Caledonia coral skeletal preservation and chronology.** The studied *D. heliopora* coral was collected in March 2015 at the

Fausse Passe de Uitoé in New Caledonia (22°17′152 S, 166°10′992 E; Fig. 1 and Supplementary Fig. 1). The methodology applied for $^{230}$Th/U-dating, $\delta^{18}$O, $\delta^{13}$C and $\delta^{11}$B isotopes measurements, chronology, climate reconstruction and statistical analyses, are described in detail in the Methods. Examination of coral skeletal material by X-radiograph imaging (Supplementary Fig. 2), powder X-ray diffraction (XRD) analysis (100% aragonite) and scanning electron microscopy (SEM; Supplementary Fig. 3), a prerequisite for coral-based palaeoclimatic work[20], attested pristine preservation. The chronology of our coral is determined by

| Coral sample | $^{238}$U ($\mu$g/g) | $^{232}$Th (ng/g) | $\delta^{234}$U (measured)[a] | $^{230}$Th/$^{238}$U (activity)[b] | $^{230}$Th/$^{232}$Th (activity) | $\delta^{234}$U (initial)[c] | Age corrected (before 2016)[d] | Age (CE) |
|---|---|---|---|---|---|---|---|---|
| 1 | 3.007 ± 0.004 | 0.054 ± 0.0001 | 146.93 ± 0.68 | 0.0025 ± 0.00002 | 413.24 ± 3.71 | 147.03 ± 0.68 | 0.231 ± 0.004 | 1785 ± 4.0 |
| 2 | 2.884 ± 0.001 | 0.080 ± 0.0001 | 146.77 ± 0.93 | 0.0034 ± 0.00004 | 366.86 ± 4.96 | 146.90 ± 0.93 | 0.313 ± 0.008 | 1704 ± 7.5 |

**Table 1 New Caledonia *Diploastrea heliopora* $^{230}$Th/U-age results**

Ages are presented as both age corrected (years before 2016) and age (CE) with the corresponding 2σ-uncertainty
[a] $\delta^{234}$U = ([$^{234}$U/$^{238}$U]$_{activity}$ −1) × 1000
[b] Decay constants are 9.1705 × 10$^{−6}$ yr$^{−1}$ for $^{230}$Th, 2.8221 × 10$^{−6}$ yr$^{−1}$ for $^{234}$U and 1.55125 × 10$^{−10}$ yr$^{−1}$ for $^{238}$U
[c] $\delta^{234}$U initial was calculated based on $^{230}$Th age (t), i.e., $\delta^{234}$U$_{initial}$ = $\delta^{234}$U$_{measured}$ × e$^{\lambda 234*t}$, and t is corrected age
[d] Age corrections were calculated using $^{230}$Th/$^{232}$Th activity ratio of 10 ± 3

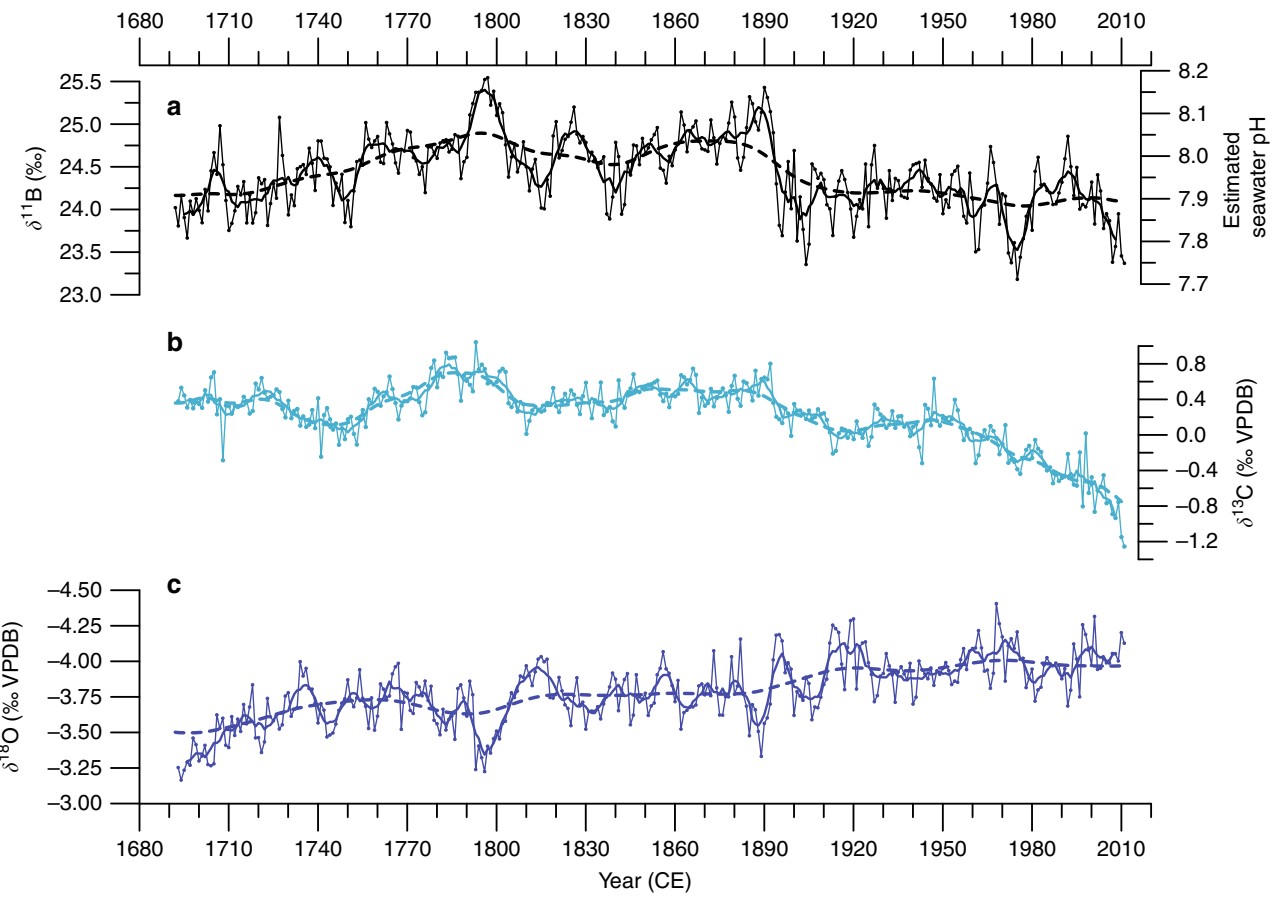

**Fig. 2** *Diploastrea heliopora* coral proxy records and $\delta^{11}$B-pH reconstruction. New Caledonia *D. heliopora* annually resolved records over the period 1689–2011 CE from precisely dated $^{230}$Th/U-age (see Methods). **a** Coral $\delta^{11}$B signature (left *y* axis) with estimated seawater pH (pH$_{SW}$) on the right *y* axis using the $\delta^{11}$B$_{SW}$ = 39.61‰ (ref. [67]) and isotopic fractionation factor ($\alpha_{[B3-B4]}$) of 1.0272 (ref. [35]) following the conversion equations of refs. [34] and [39] (see Methods for Eqs. (1) and (2)). **b** Coral $\delta^{13}$C (‰ VPDB) and **c** $\delta^{18}$O (‰ VPDB) ratios. To highlight the interannual variability, we applied a 7-year smoothing to the records (bold lines). The long-term secular trends of each time series (dashed lines) indicate the long-term changes and analysed by Bayesian Change Point Analysis algorithm[72], revealing significant times of emergence (see Methods)

density banding counting (Supplementary Figs. 2 and 4) and verified by independent high-precision $^{230}$Th/U-dating in the older part of the skeleton (Table. 1). We established that our *D. heliopora* coral colony grew continuously over the period 1689–2014 CE at a rate of $2.68 \pm 0.64$ mm per year, placing the proxy records and reconstructions precisely in the termination of the Little Ice Age (LIA) during the solar Maunder Minimum, the onset of industrial era from the late 1700s, and through the modern climate regime of today.

**$\delta^{18}$O calibration and SST reconstruction**. We generated annually resolved records of coral $\delta^{11}$B, $\delta^{13}$C and $\delta^{18}$O values for a total of 323 continuous years over the Common Era (1689–2011 CE; Fig. 2 and Supplementary Fig. 5). On seasonal to sub-seasonal timescales, physiological processes such as photosynthesis, respiration and heterotrophy influenced by metabolic fractionation clearly affected coral $\delta^{13}$C levels[21,22]. However, over longer-term centennial timescales, we observed coral $\delta^{13}$C depletion towards more negative values that can be attributed to the oceanic $^{13}$C Suess effect[23] and may be used as an indicator of changes in ocean carbon content or the DIC in seawater[24] and anthropogenic $CO_2$ uptake[25]. Coral $\delta^{18}$O signature is one of the most frequently used coral climate proxies because it is influenced by the combination of temperature-dependent isotope fractionation and the $\delta^{18}$O values of seawater ($\delta^{18}O_{sw}$)[26]. Such value is directly related to sea surface salinity (SSS) responding to evaporation and freshwater input. Previous studies from the Pacific have demonstrated that coral $\delta^{18}$O variations over multiple timescales represent changes of SST, SSS, precipitation, or a mixture of those parameters depending on the environmental setting[19,27,28].

The calibration of our *D. heliopora* coral $\delta^{18}$O ratio to SST from New Caledonia was completed using annual gridded SST from the National Oceanic and Atmospheric Administration (NOAA), National Centers for Environmental Information (NCEI) and Optimum Interpolation SST version 2 (OI-SST)[29]. A 4-grid averaged OI-SST (Supplementary Table 1) was chosen because coral $\delta^{18}$O-based SST is not only a local archive, but is in fact encapsulating a larger regional signal[19,27]. The least squares linear regression calibration of annually averaged $\delta^{18}$O to the 4-grid OI-SST is significant ($p < 0.001$). Even though coral $\delta^{18}$O record is a combination of both SST and the $\delta^{18}O_{sw}$, the coral $\delta^{18}$O-SST sensitivity of $-0.24‰$ $C^{-1}$ is in agreement to the typically described relationship across the Pacific even with the use of annual resolution sampling (Supplementary Fig. 6a). *D. heliopora* species-specific calibrations using in situ and gridded or satellite-based SST indicated a SST-sensitivity relationship ranging from $-0.16$ to $-0.22‰$ per °C for sub-seasonal calibrations[30–32], which is similar to our finding.

We cross-verified the accuracy of our calibrated $\delta^{18}$O-SST record and chronology across the overlapping period (1860–2011 CE) to the NOAA NCEI Extended Reconstructed SST version 4 (ERSST)[33]. A 4-grid regional annual average SST for ERSST (Supplementary Table 1) was compiled and compared as a moving average, which increased the signal-to-noise ratio to track the prominent interannual to decadal-interdecadal changes. The calculated running correlation between our coral $\delta^{18}$O-SST and ERSST yielded significant ($p < 0.01$) and high correlation coefficients between the most recent period of the record with the most accurate instrumental SST period (1960–2011 CE; Supplementary Fig. 6b). This considerable coherence in the late twentieth century (Supplementary Fig. 6b) corresponds to the period when the number of SST observations is higher with highest degree of confidence (Supplementary Fig. 7). The misfit between coral $\delta^{18}$O-SST record and the gridded SST is greater in the middle of the twentieth century, a period when the

number of International Comprehensive Ocean-Atmosphere Data Set (ICOADS; http://icoads.noaa.gov/products.html) observations in the gridded SST timeseries was low. The observed lack of significant coherence over the mid-twentieth century in our analysis and verification may be reflecting the inadequacies of the gridded instrumental SST product. Nonetheless, the result over the verification period provides confidence in our chronology and the potential of the coral $\delta^{18}$O-SST proxy although it is likely that the contribution of $\delta^{18}O_{sw}$ is of significance. Thus, our $\delta^{18}$O record from New Caledonia suggests a strong secular warming and or freshening trend consistent with many coral records from the western Pacific with similar timing for the onset of warming in the mid-nineteenth century[19] (Fig. 2).

**Multicentennial coral $\delta^{11}$B pH variability**. To reconstruct seawater carbonate chemistry changes and potential modern ocean acidification from anthropogenic $CO_2$ uptake at the sea surface, we utilised the $\delta^{11}$B composition in our coral. In seawater, the relative abundance of the two aqueous boron species (boric acid and borate) as well as their isotopic composition are pH dependent[34] with a constant fractionation factor between the two aqueous boron species[35]. On the assumption that marine carbonates reflect the isotopic composition of seawater borate[36], the $\delta^{11}$B signature of coral aragonite skeleton has been shown to record seawater pH[12,37]. Possible intra-colony differences of $\delta^{11}$B signature of coral aragonite skeleton were taken into account as studies of sub-seasonally resolved *D. heliopora* coral $\delta^{18}$O and Sr/Ca ratios have shown offsets between skeletal materials (columellar and septal) and different coral polyps[30–32]. However, at the annual resolution, our test across multiple coral polyps revealed that skeletal material differences are inconsequential for $\delta^{11}$B ratios (Supplementary Fig. 8). The three replicated sections of 20-year periods (1935–1955, 1856–1876 and 1768–1788 CE) were reproducible with negligible intra-colony offset that were within the analytical errors between the different polyps and sampling paths (Supplementary Fig. 8). The large amount of coral skeletal material used for $\delta^{11}$B analysis (~50 mg) compared to traditional $\delta^{18}$O and $\delta^{13}$C analyses makes it statistically likely that the sample comprises all different skeletal structures (columellar and septal) and is thus representative of the whole coral skeleton. Our method demonstrates the reliability and reproducibility of annual $\delta^{11}$B ratio results from a new south Pacific coral species (*D. heliopora*) with larger skeletal structure than the commonly used *Porites* coral genus.

Careful considerations were taken to convert our coral $\delta^{11}$B signature to coral internal pH as studies indicated differences between empirically and theoretically determined fractionation factors for isotope exchange between boric acid and borate in seawater[35]. Furthermore, results based on a variety of coral species across temperate and tropical regions indicate consistent difference between coral internal calcification pH by approximately one-half of ambient seawater pH[38,39]. However, recent results also indicated possible modifications of coral internal pH due to external factors and species-specific processes[40,41]. Thus, we estimated the changes in seawater pH at New Caledonia using a moderate general calibration conversion[39] (Fig. 2). To be conservative with our conversion, a calibration based on a similar massive coral species (*Porites* spp.) to the one used in this study (*D. heliopora*) was chosen allowing for the establishment of biological and seawater pH linkage to climate-driven patterns. We focused our discussion on the relative pH changes instead of the absolute pH values because rigorous palaeo-pH reconstructions from coral $\delta^{11}$B signature require species-specific calibrations with robust quantification of physiological processes (vital effects). Nevertheless, independently from a possible offset in

our reconstructed pH using this calibration conversion, our observed trend and interannual to decadal variability from the measurements along the core will remain.

Between the independent proxies of $\delta^{11}B$, $\delta^{13}C$ and $\delta^{18}O$, significant relationships were found. The significant correlation between $\delta^{11}B$ and $\delta^{13}C$ ($R = 0.61$; $p < 0.01$, $n = 319$; Supplementary Fig. 9a) indicates a strong linear relationship indirectly measuring the amount of atmospheric $CO_2$ that is forced by fossil fuel emissions and subsequently taken up by the ocean as reflected in the changing oceanic DIC content. Moreover, the correlation between $\delta^{11}B$ and $\delta^{18}O$ is also significant ($R = 0.42$, $p < 0.01$, $n = 319$; Supplementary Fig. 9b) but indicates a weaker linear dependence because coral $\delta^{18}O$ record integrates both SST and the $\delta^{18}O_{sw}$ (water balance at the sea surface). Obvious perturbations outside of the regression line were observed and distinguished the most recent samples related to anthropogenic $CO_2$ emissions in the atmosphere from those related to the older preindustrial era period (Supplementary Fig. 9).

## Discussion

As the ocean is one of the major global sinks of anthropogenic $CO_2$ emission, the incursion of isotopically light carbon ($^{12}C$)

from the burning of fossil fuel into the ocean has caused considerable decrease in the $\delta^{13}C$ of seawater DIC, known as the $^{13}C$ Suess effect. The $\delta^{13}C$ ratio depletion trend in atmospheric $CO_2$ at five Pacific stations[42], the Law Dome ice core record[43], and archived in numerous Pacific coral records (Supplementary Tables 2 and 3) are remarkably indistinguishable (Fig. 3). Since 1978, both instrumental and proxy records indicate a consistent average of −0.027‰ per year decrease in $\delta^{13}C$ (Supplementary Table 3). We can easily rule out wind-driven upwelling influence on our $\delta^{13}C$ record because its dominant effects are experienced on daily to weekly timescales[44] and not at annual to longer timescales. The geochemical variations observed in our coral cannot be attributed to the linear growth rate because the vertical extension rates remained relatively constant (2.68 ± 0.64 mm per year) throughout the interval of this study. For this particular coral genus, similar vertical extension rates have been reported throughout the Pacific[30,31].

This long-term secular trend towards lower $\delta^{13}C$ values is neither related to tropical Pacific solar radiative forcing[45] nor tropical volcanic forcing[46] (Fig. 3 and Supplementary Table 4). The emissions of $CO_2$ from volcanic eruptions since 1750 (Supplementary Table 4) are mostly inconsequential for climate on centennial to millennial timescales as it is at least 100 times

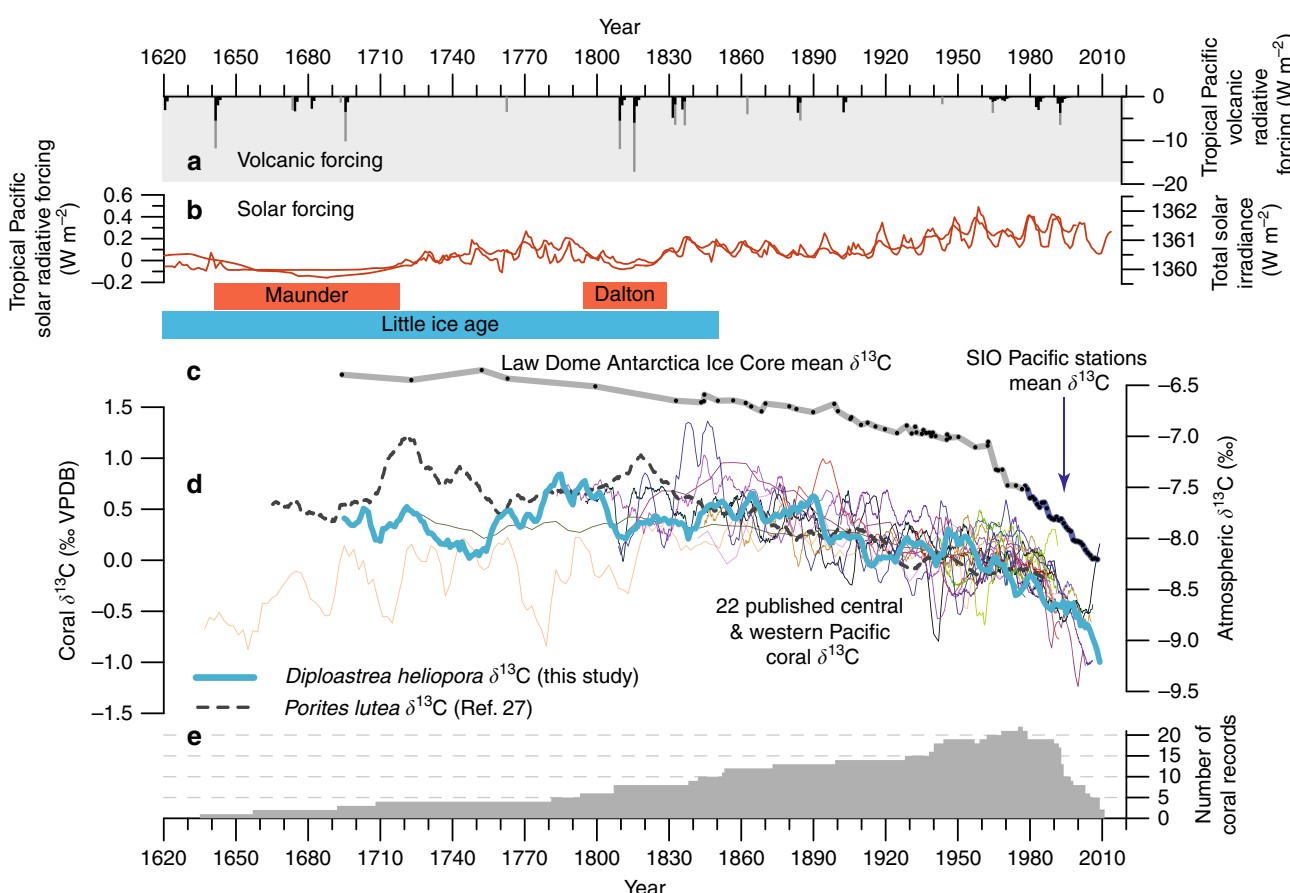

**Fig. 3** Tropical Pacific coral $\delta^{13}C$ records and external forcings. **a** The major tropical Pacific volcanic eruptions (Supplementary Table 4) and total radiative forcing from volcanic events and **b** solar variability[45,46] indicate a lack of direct influence on the atmospheric and coral $\delta^{13}C$ values. **c** Atmospheric $\delta^{13}C$ values recorded at observation stations in the Pacific[42] (blue; Supplementary Table 3) and in the Law Dome Antarctica ice core (grey; mean $\delta^{13}C$ record[43]) indicate strong coherence and similar timing of significant secular depletion of $\delta^{13}C$ since the industrial revolution. **d** Independent records of 22 published coral $\delta^{13}C$ signature from the central and western Pacific (Supplementary Table 2) also recorded notable acceleration in long-term $\delta^{13}C$ depletion due to increase in anthropogenic $CO_2$ output and oceanic uptake. **e** The individual coral time series vary in their native temporal resolution (monthly to 5-year average) and are all downscaled to 5-year moving means to facilitate comparison. Due to known inter-colony and inter-species offsets in coral $\delta^{13}C$ records, we removed the mean $\delta^{13}C$ value from each individual record over the twentieth century to centre the records

smaller than anthropogenic emissions[47] . These consistent secular trends from instrumental measurements and the rapid rate of coral $\delta^{13}C$ depletion found at New Caledonia (−0.024‰ per year; Supplementary Table 3) recording the $\delta^{13}C$ of seawater DIC over the same period (1978–2011; Fig. 3 and Supplementary Table 3) demonstrate the significant absorption of anthropogenic $CO_2$

emission by the oceans. The pronounced long-term depletion since 1843 by at least 1.0‰ in our *D. heliopora* $\delta^{13}C$ record is in agreement with the nearest New Caledonia *Porites* sp. $\delta^{13}C$ record that found a depletion of 0.9‰ since the late nineteenth century[27], confirming the modern uptake of anthropogenic atmospheric $CO_2$ by the tropical oceans.

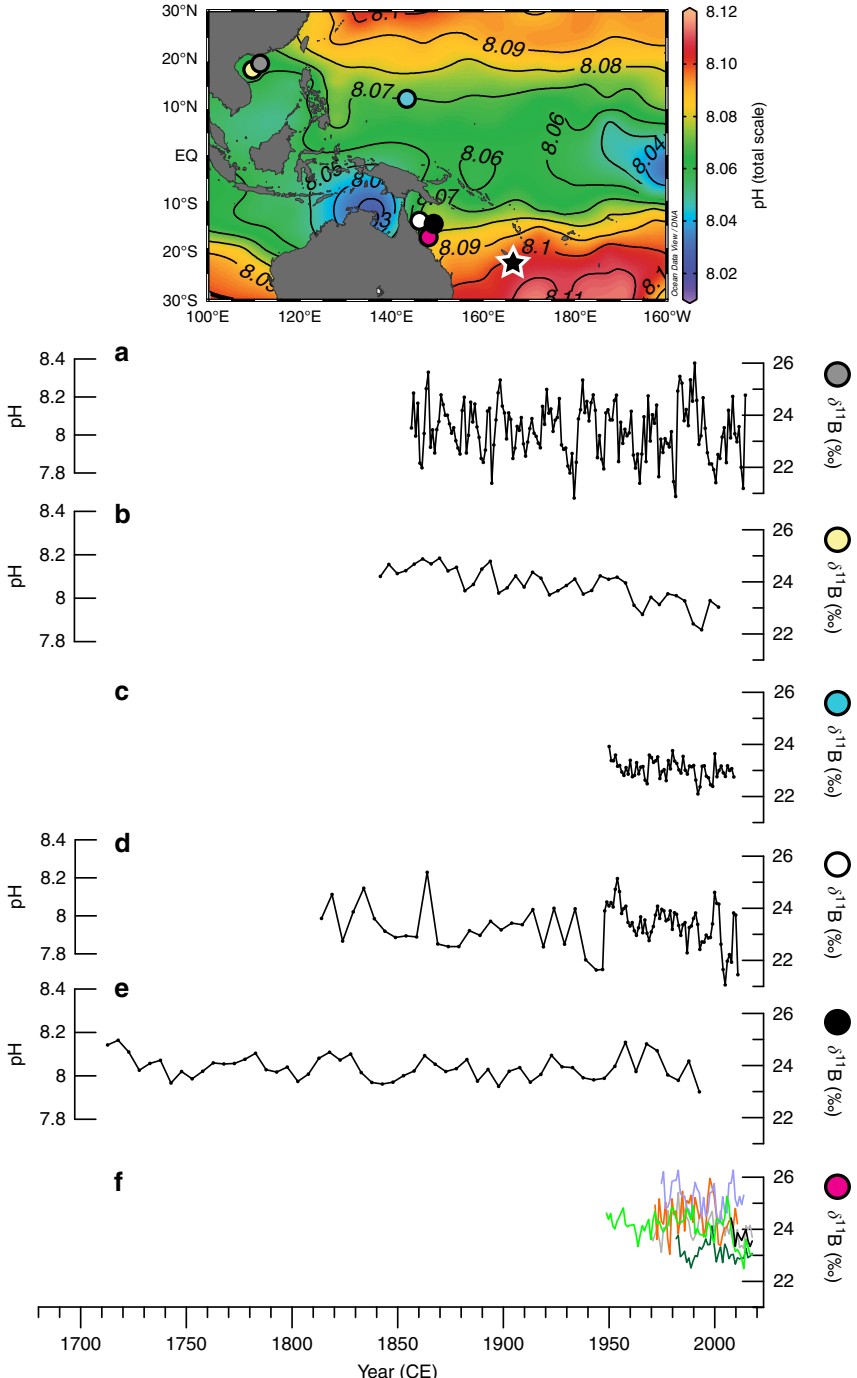

**Fig. 4** Pacific Ocean modern coral $\delta^{11}B$ and pH reconstructions. The published and publicly available coral $\delta^{11}B$ and pH reconstructions from the Pacific Ocean are shown in the annual gridded mean estimated pH map[10] and listed on Supplementary Table 5. Each individual symbol corresponds to each respective record's research location. The records are presented relative to the location of the cores from the north to the south. The locations include two records from **a**, **b** the South China Sea[13,14], **c** one from Guam[17], and **d**, **e**, **f** three sites in the Great Barrier Reef (GBR)[12,15,16]. The records from **f** Havannah Island, GBR[16] (magenta dot) are the results of six individual *Porites* coral colonies. All $\delta^{11}B$ records are shown in their native temporal resolution as given in the original publication. The accompanying pH reconstructions are indicated on a secondary y axis if provided by the original publication and were not converted to seawater pH based on our methodology (see Methods). Map is produced using Ocean Data View ver. 4.7.4 (ref. [74]) (http://odv.awi.de) and modified manually

Over the period 1689–2011 CE, our unique $\delta^{11}B$ record from New Caledonia (Supplementary Fig. 1) provides the first pH variability reconstruction reflecting open ocean conditions as the currently available limited reconstructions are derived from enclosed reef sites and marginal seas in the Pacific[12–17] (Fig. 4). On both interannual and decadal-interdecadal timescales, the annually resolved $\delta^{11}B$ values as a proxy of pH show pronounced oscillations (Fig. 2). These high-amplitude $\delta^{11}B$ or estimated pH variations in the surface waters in the South Pacific indicate large interannual changes of up to 0.8‰ equating to a 0.16 pH unit change in seawater (Fig. 2). The extremes of interdecadal fluctuations are smaller than those observed at the higher interannual frequencies with an average change of 0.4‰ in $\delta^{11}B$ or 0.08 pH unit in seawater. The swings between 0.08 and 0.16 pH units in South Pacific pH over interannual to interdecadal timescales are not approaching the high benchmark of 0.4 pH units for end-of-century emission scenarios as predicted by model simulations[1]. The full amplitude of our $\delta^{11}B$ and reconstructed seawater pH changes are well-within the magnitude of previously reported coral-based pH reconstructions[12,13,17] regardless of the absolute-scale pH reconstruction, such as the 0.3 pH units of interdecadal variability found at Flinders Reef, GBR[12] (Figs. 2, 4 and Supplementary Table 5). Moreover, studies from the South China Sea[14] and the GBR[15] recorded large interannual to interdecadal fluctuations (0.6–0.7 pH units) that are even greater than the 0.4 pH units predicted for the end-of-the-century (Fig. 4). The large pH discrepancy of the near-shore records from the South China Sea[14] and GRB[15] may indicate the large spatial variability of pH and possible coastal riverine influence[16] or local reef effect as compared to our open-ocean record.

In contrast to our coral-based pH reconstruction, hindcast model simulations of historical pH from the Pacific are unable to reproduce any significant modes of longer-term lower frequency variability[48]. Questions arise on the reliability of these model reconstructions and simulations based solely on Pacific SST and atmospheric $CO_2$ for the pre-industrial period. In addition, calculated pH reconstructions based on in situ $pCO_2$ measurements of the past decades also lack significant low frequency variability[49,50]. It is likely that the underlying observation periods used for these reconstructions and model simulations are insufficient to capture the longer timescale variability that is taking place in the Pacific Ocean. Thus, our coral $\delta^{11}B$ record appears to document the longer timescale pH changes in the Pacific that cannot be archived from the marginal seas or expressed in reanalysis studies and model simulations.

Our annually resolved $\delta^{13}C$ also reveals significant interannual and interdecadal variability of up to 0.4–0.5‰. The temporal variability in $\delta^{13}C$, which can be mainly attributed to changes in seawater DIC, is nearly synchronous to those of $\delta^{11}B$. The lower frequency variability and the magnitude of change is consistent with that found in tropical shallow water corals from both the Atlantic and the Pacific oceans[25,27] (Fig. 3). Despite the independent nature and behaviour of both proxies ($\delta^{11}B$ and $\delta^{13}C$), 37% of the variability in $\delta^{11}B$ can be explained by $\delta^{13}C$ from correlation analysis (Supplementary Fig. 9). This interrelationship over interannual to interdecadal timescales thus indicates a possible common driver that contributed to the changes of both proxies. We suggest that the relationship between coral $\delta^{11}B$ and $\delta^{13}C$ records at these timescales illustrates consistent and considerable marine carbonate chemistry changes in the surface ocean of the South Pacific. The shallow water mixing and uptake/release of $CO_2$ is mostly controlled by the temperature-dependent solubility of $CO_2$ over interannual timescales and it is likely that surface wind strength and hydrodynamic changes of the Pacific Ocean is the dominant process over longer decadal-interdecadal timescales. Working in combination, this dominant process of the

Pacific may dampen the interannual signature of the shallow water mixing and $CO_2$ uptake/release changes.

Another striking result from our coral reconstruction is the long-term decreasing secular trend of the $\delta^{11}B$ signature from 24.72 to 24.09‰ over the most recent century. This major depletion corresponds to a consistent decreasing pH of 0.12 pH units since the late nineteenth century. The trend of decreasing $\delta^{11}B$, as well as the marked increasing uptake of $CO_2$ depicted in the lowering of coral $\delta^{13}C$ values, substantially matches the expected trend for the greater Pacific (Figs. 2 and 3). Bayesian Change-Point Analysis (CPA; see Methods for details) determined the date of the initiation of this decreasing trend for our $\delta^{11}B$ and $\delta^{13}C$ records. For $\delta^{11}B$, the timing of significant modern OA emergence occurred at 1886, between the confidence interval of 1881–1891, and after the significant depletion emergence of $\delta^{13}C$ in 1843 (confidence interval: 1829–1850; Figs. 2 and 3). The timing of this event emergence between $\delta^{11}B$ and $\delta^{13}C$ is thus lagged by ~40 years, which indicates the link of the slow ocean pH and carbonate chemistry changes following the uptake of atmospheric $CO_2$.

Before this onset of modern anthropogenic-driven OA, a coral $\delta^{11}B$ pH maximum was reached in the late 1790s (Fig. 2). The increase in $\delta^{11}B$ at the centennial scale from the sixteenth to the seventeenth century appears to be decoupled from coral $\delta^{13}C$ trend as the seawater DIC signature remained relatively level. We observed that the progressive increase in $\delta^{11}B$ pH during this time period (1701–1761 CE) coincided with changes in temperature most likely linked to the termination of the LIA, a period that was documented to be 1.4 °C cooler at New Caledonia[27,51]. The end of this period was however characterised by a maximum $\delta^{11}B$ pH that is contrary to the recorded maximum $\delta^{18}O$ enrichment (cooler and/or more saline conditions; Fig. 2). It is possible that as a consequence to the termination of the LIA, a redistribution of water masses occurred near New Caledonia, which experienced an intrusion of cooler and/or more saline water to the region from the enhanced subtropical countercurrent[52] (Supplementary Fig. 10). These connections point to the substantial linked behaviour of pH, temperature and salinity, at the longer-term centennial timescales.

The coral-based high-amplitude pH variations occurring on interannual to decadal timescales in New Caledonia are accompanied by strong coherent SST or $\delta^{18}O_{sw}$ variations for most time intervals inferred from the coral $\delta^{18}O$ proxy (Fig. 2). In general, coral $\delta^{11}B$ and $\delta^{18}O$ records display anti-phase coherency with higher SST concomitant with more acidic conditions. Spectral analyses reveal significant variability for both $\delta^{11}B$ and $\delta^{18}O$ coral records on interannual (3–7 years) and decadal-interdecadal (11–41 years) timescales (Supplementary Fig. 11). These frequencies are typical of the well-documented ENSO and IPO climate modes of variability as observed in both instrumental[53] and coral proxy records[27,54] in the South Pacific. Instrumental observations suggest that ENSO events are the major control mechanisms for seawater $pCO_2$ variation in the equatorial Pacific on interannual to decadal timescales[50]. During most El Niño (EN) events, the $\delta^{11}B$ signature decreases from the preceding year translating into a surface seawater decrease in pH, and the opposite occurs during La Niña (LN) events with an increase in $\delta^{11}B$ signature from the preceding year (increase in surface pH; Fig. 5 and Table 2).

The most recent ENSO events indicate possible changes of up to ±0.35‰ in coral $\delta^{11}B$ (Fig. 5 and Table 2), which can equate to ±0.07 pH units change in seawater pH for a single event under the most severe conditions. An additional important observation is that not all severe or very severe EN events result in the same magnitude of $\delta^{11}B$ depletion. Some moderate EN events (based on ERSST[33] amplitude; Oceanic Niño Index) displayed more

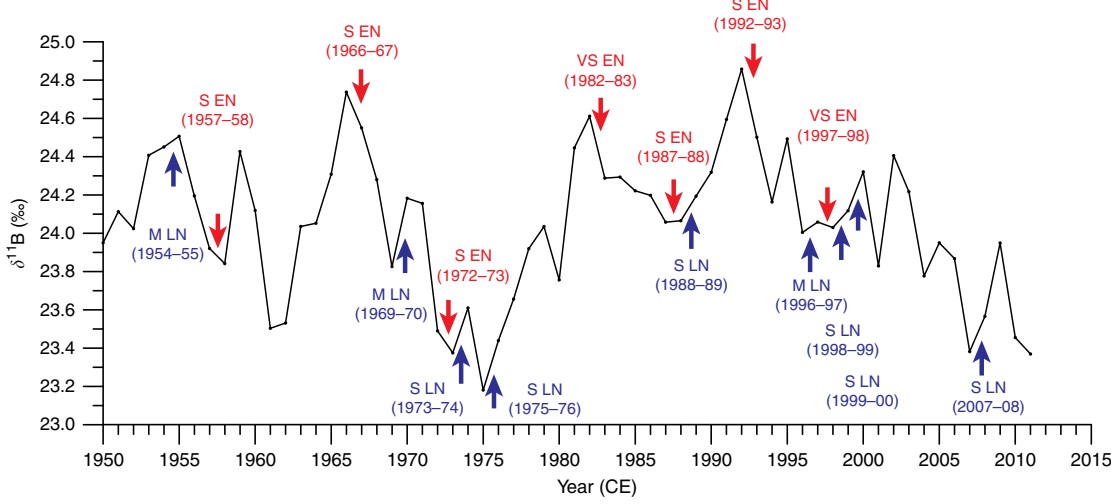

**Fig. 5** New Caledonia coral $\delta^{11}$B and ENSO events. The New Caledonia coral $\delta^{11}$B record from 1950–2011 CE with recent ENSO events identified (Table 2). The severity of the listed El Niño (EN) or La Niña (LN) events are identified by the Oceanic Niño Index (ONI) from the 3-month running mean SST anomaly that is above or below the 0.5 °C threshold for a period of at least 5 months in the Niño 3.4 region based on ERSST version 4 (ref. [33]). The threshold of ENSO event is broken down into categories of moderate -M- (±1.0–1.4 °C), severe -S- (±1.5–2.0 °C) and very severe -VS- (>±2.0 °C)

| Table 2 ENSO events from Oceanic Niño Index | | | | | | | |
|---|---|---|---|---|---|---|---|
| **El Niño** | | | | **La Niña** | | | |
| **Severe (S)** | $\delta^{11}$**B change (‰)** | **Very Severe (VS)** | $\delta^{11}$**B change (‰)** | **Moderate (M)** | $\delta^{11}$**B change (‰)** | **Severe (S)** | $\delta^{11}$**B change (‰)** |
| 1957–58 | −0.079 | 1982–83 | −0.323 | 1954–55 | +0.056 | 1973–74 | +0.235 |
| 1966–67 | −0.187 | 1997–98 | −0.029 | 1969–70 | +0.357 | 1975–76 | +0.259 |
| 1972–73 | −0.115 | | | 1996–97 | +0.054 | 1988–89 | +0.128 |
| 1987–88 | 0.007 | | | | | 1998–99 | +0.088 |
| 1992–93 | −0.356 | | | | | 1999–00 | +0.293 |
| | | | | | | 2007–08 | +0.184 |

Historical ENSO events, El Niño (warm) and La Niña (cold) from the Oceanic Niño Index (ONI) calculated as the 3-month running mean SST anomaly in the Niño 3.4 region (ERSST ver. 4[33]) that is above or below the 0.5 °C threshold for a period of at least 5 months. The threshold is further broken down into categories of weak (±0.5–0.9 °C), moderate (±1.0–1.4 °C), severe (±1.5–2.0 °C) and very severe (>±2.0 °C)

pronounced depletion of $\delta^{11}$B than severe EN events and the very severe EN of 1997–98. Severe LN events are consistent with enrichment of $\delta^{11}$B indicating an increase in pH (Fig. 5 and Table 2). In addition, periods of more 'active' ENSO activity[55] (1890s–1910s) are recorded in our coral as a series of major depletions indicating decreases in pH in New Caledonia. The fluctuations in our coral-based $\delta^{11}$B pH thus reflect the highly dependent nature of ocean $CO_2$ uptake across the air–sea surface interface following the pacing of ENSO.

A remarkable atmospheric teleconnection is witnessed by the compelling coherence and in-phase relationship of our New Caledonia reconstructed pH from the South Pacific to the most ENSO representative region, the Niño 3.4 region SST anomaly record[56], more than 6000 km away (Fig. 6). This decadal coherence at the 11–12-year frequency may be part of the combined modes of climate variability (e.g., ENSO and IPO) exhibiting the decadal modulation of interannual ENSO. This result demonstrates that the long-term changes in pH observed at New Caledonia are neither local nor specific to the South Pacific and are in fact a Pacific basin-wide manifestation (Fig. 6). Moreover, recent IPO phase shifts of Pacific-wide SST[53] are clearly observed in our $\delta^{11}$B record corresponding to the years 1945, 1977 and 1999. The changes in SST affect the solubility of $CO_2$ and can partially explain this lower frequency variability at 11–12 years especially in the tropical and subtropical regions[57]. We argue that this result

is mainly due to the confluence of interannual-decadal SST changes in the Pacific and the changing surface wind strength that modulated the uptake of $CO_2$. More frequent occurrence of equatorial westerlies during rapid warming periods involve weakened trade winds[58], supporting the connection between South Pacific pH and tropical Pacific SST variability. Therefore, the pronounced and coherent high-amplitude variability in SST and pH over ENSO and decadal-interdecadal timescales indicated by our coral suggests a dominant role of surface wind strength (trade winds and westerlies) changes and zonal advection of surface heat in the Pacific Ocean as the primary cause and mechanism of surface ocean carbonate chemistry mixing and ultimately pH variability.

Cross-wavelets coherence analysis detecting frequency and relative phase relationship between our proxy records indicates significant non-stationary relationships between pH and SST/$\delta^{18}O_{sw}$ of the South Pacific (Fig. 7). Dominant low-frequency coherence at the decadal-interdecadal variability (~40 years) was observed between our coral $\delta^{11}$B and $\delta^{13}$C records from the beginning of the record up to the emergence of acidification (mid-1800s). This relationship then shifted to a more prominent higher-frequency variability of 7 years (ENSO-related) to 16 years (Fig. 7). Similarly, the relationship between $\delta^{11}$B and $\delta^{18}$O exhibits this significant transposition to higher-frequency variability evolving from decadal-interdecadal frequency (16–32 years) to

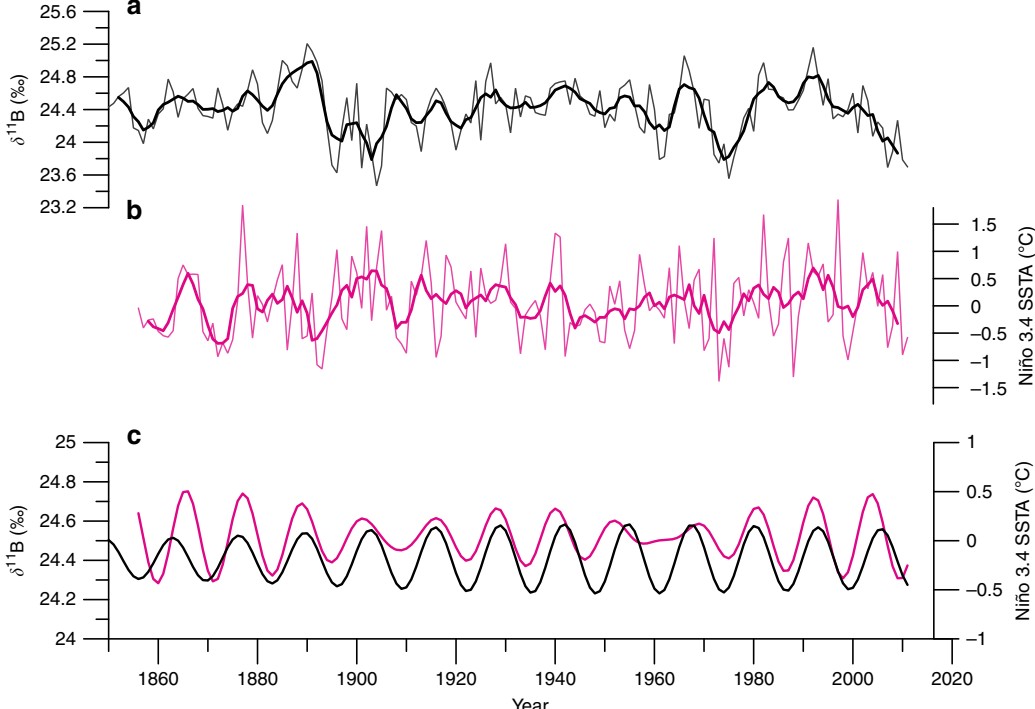

**Fig. 6** New Caledonia coral $\delta^{11}$B and Niño 3.4 SSTA spectral coherence. **a** Detrended $\delta^{11}$B time series with smoothed 5-year running mean in bold indicating interannual variability. **b** Detrended Niño Region 3.4 SST anomalies (Niño 3.4 SSTA)[29] with smoothed 5-year running mean in bold indicating interannual variability. **c** The extracted decadal mode (11–12 years) from Singular Spectrum Analysis of the New Caledonia *D. heliopora* $\delta^{11}$B (black) and the Niño 3.4 SSTA (magenta) indicating coherent and synchronous decadal variability across the Pacific Ocean, from the southwestern Pacific to the far eastern Pacific

the ENSO frequency (4–8 years) in the last 100 years. Such observations are consistent with Central Pacific and Australian coral-based ENSO studies, where ENSO variance is significantly higher in the twentieth century than in any other time since the late Holocene[59]. We conclude that the non-stationary behaviour of long-term pH and SST variability observed in New Caledonia, which shifted to more pronounced interannual fluctuations after ~1850 cannot be due to simple local ocean circulation changes. This prominent shift rather suggests a likely anthropogenic influence with strong $CO_2$ uptake by the ocean because the significant trend changes and emergence timing coincides with these frequency evolution in the post-industrial era.

In summary, our reconstructed pH record from the South Pacific highlights that marine carbonate chemistry in the South Pacific is highly variable with pronounced high-amplitude pH oscillations since 1689 CE. We observed a significant emergence of secular decreasing pH trend since 1886 together with the dramatic depletion of oceanic $\delta^{13}$C that is related to the atmospheric $^{13}$C Suess Effect caused by the uptake of anthropogenic $CO_2$ by the ocean. We argue that the onset of this modern acidification in the mid-1800s and the temporal coherence of $\delta^{11}$B pH and $\delta^{13}$C or marine carbonate chemistry changes are coupled to SST variability and have been consistently occurring for the last 300 years. The temporal coherence of the proxy records also has significantly changed in frequency since the emergence of the modern acidification. We implicate the changing strength of the westerlies and trade winds as important drivers forcing the pH and SST changes across the greater Pacific under possible decadal-enhanced ENSO variability (11–12 years). The changing surface ocean condition is a distinct process that modified the carbonate chemistry of the Pacific over the last three centuries with recent changes likely dominated by anthropogenic influence. Our results provide a new detailed perspective on the

changes in modern OA that was likely influenced by the inseparable combination of the changing surface winds strength (trade winds and westerlies) and ENSO-related SST conditions driving the uptake of anthropogenic $CO_2$ in the South Pacific.

## Methods

**Coral sampling and skeletal verification**. In March 2015, a *D. heliopora* coral was cored in 7–8 m water depth at the Fausse Passe de Uitoé, New Caledonia in the southwestern Pacific (22°17′152 S, 166°10′992 E; Fig. 1 and Supplementary Fig. 1). The 7-cm diameter core was 1.3 m in total length and returned to Institut de Recherche pour le Développement (IRD) in Bondy, France in August 2015 for processing and analyses. The coral core was halved down the vertical central growth axis and identical ca. 7 mm slabs were removed from each core half. Each coral slab was cleaned in an ultrasonic bath with 18.2 MΩ milli-Q water for 30 min and air-dried at room temperature. X-radiographs of the coral micro-sampling slab were completed at the L'Institut Mutualiste Montsouris Paris (Supplementary Fig. 2).

Five subsamples from the coral core were systematically removed down the length of the coral core. Each subsample was further divided into individual portions for powder X-ray diffraction (XRD) and SEM analyses at the IRD, Laboratoire d'Océanographie et du Climat (LOCEAN) on the Analytical Platform ALYSES (IRD-UPMC) at Bondy, France. Approximately 500 mg of crushed coral powder (<20 μm) for all subsamples were analysed on a PANalytical X'Pert Powder X-ray Diffractometer that revealed 100% aragonite. Verification of the coral's suitability for palaeoclimate research by representative SEM analysis was conducted on a Zeiss EVO LS15 SEM. The samples were carbon coated and analytical results demonstrated good preservation of primary porosity, clear dissepiments and well-defined centres of calcification (Supplementary Fig. 3). Minor evidence and presence of secondary aragonite away from the regions of micro-sampling indicated alterations that are well within the limits of minor alterations in corals for palaeoclimate reconstruction studies[20].

Based on the X-radiograph positives of the coral slab with well-defined annual skeletal density banding, the chronology and age model of the coral colony was developed based on band counting and then verified by $^{230}$Th/U-ages. The most recent 3 complete years of growth (2011–2014 CE) were omitted because of the presence of organic tissue layer remnants (~9–10 mm thick), which would have biased the geochemical results. Similarly, the coral section older than 1689 CE was avoided because the coral growth departed from the central vertical axis.

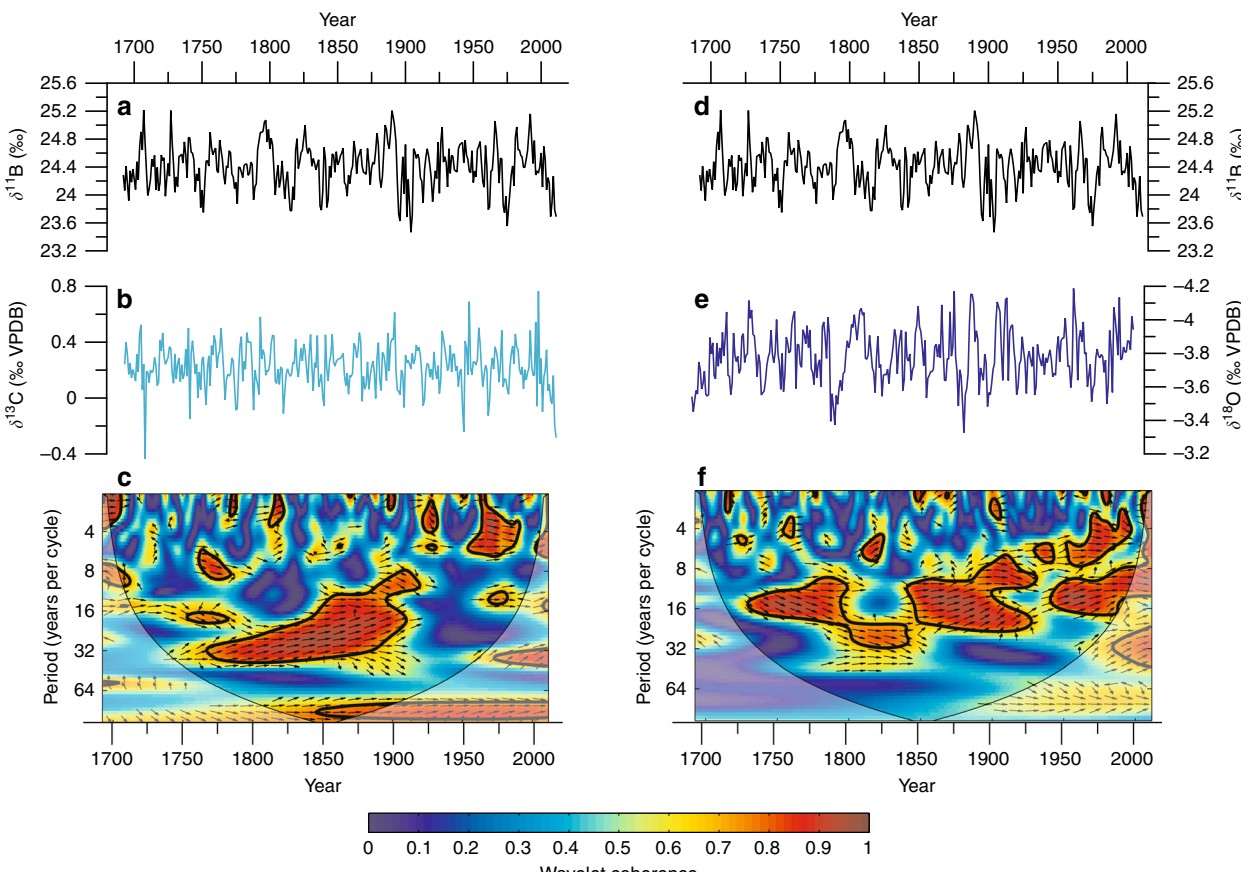

**Fig. 7** Cross-wavelet results of proxy variability in the South Pacific. **a** The coral $\delta^{11}B$ (secular trend removed; trend shown in Fig. 2) compared with **b** $\delta^{13}C$ with **c** cross-wavelet coherence analysis between the two time series with black contour lines enclosing time-periodicity regions with significant concentrations of coherence. Arrows to the right indicate positive coherence. **d** Coral $\delta^{11}B$ compared with **e** $\delta^{18}O$ with **f** cross-wavelet coherence results indicating a significant shift in periodicity concentration from the earlier part of the record with longer-term decadal-interdecadal variability to the most recent years that are dominated by interannual changes

The micro-sampling strategy is similar to the methods used in many coral-based palaeoceanography studies and for this coral genus with a slow milling procedure to minimise sample flaking[31]. Milling was completed with a Proxxon mill for coral powder collection at predetermined annual sampling intervals based on the X-radiographs (Supplementary Fig. 4). The sampling transect on the coral slab remained predominantly on a single coral polyp (1 cm in diameter) where possible (Supplementary Fig. 2). The distance for annual sampling was based on density banding (ca. 3 mm) with a width ca. 1 cm that corresponds to one polyp and contains a homogenised mixture of septal and columellar skeletal materials.

**$^{230}$Th/U-age determination.** For chronology verification, two samples were removed from the oldest sections of the coral as close to the sampling transect as possible. Due to sample size requirement for analysis, each sample contained ca. 4–5 years of coral growth (Supplementary Fig. 2). $^{230}$Th/U-age determinations of the corals were completed on a multi-collector inductively coupled plasma mass spectrometer (Thermo Neptune Plus MC-ICP-MS) on the Analytical Platform PANOPLY (LSCE-GEOPS), at the Laboratoire des Sciences du Climat et de l'Environnement (LSCE), Centre National de la Recherche Scientifique (CNRS), Gif-sur-Yvette, France. Details of the procedure (chemical preparation and analysis) can be found in ref. [60]. The initial ($^{234}$U/$^{238}$U) activity ratios of samples are in agreement with the initial ($^{234}$U/$^{238}$U) activity ratio of modern seawater (i.e., 146 ± 0.8‰; ref. [61]). The ages presented herein (Table. 1) are considered reliable and criteria for the reliability of the coral $^{230}$Th/U-age are fulfilled within their $2\sigma$-error or 95% confidence interval[61].

**Coral stable isotope ratios analyses ($\delta^{18}O$ and $\delta^{13}C$).** Coral powder samples for $\delta^{18}O$ and $\delta^{13}C$ ratios analyses were collected at annual resolution and individually homogenised using mortar and pestle at the same time as the samples used for $\delta^{11}B$ analyses. About 100 µg of powder from each sample was analysed for $\delta^{18}O$ and $\delta^{13}C$ ratios using a VG Optima Stable Isotope-Ratio Mass Spectrometer on the Analytical Platform PANOPLY (LSCE-GEOPS), at the LSCE, CNRS, Gif-sur-Yvette, France. The isotope ratios are reported in ‰ deviation relative to the

Vienna Pee Dee Belemnite (V-PDB). Long-term analytical precision based on repeated measurements of an in-house marble carbonate standard were ±0.04‰ (±1$\sigma$ standard deviation, SD; $n = 116$) for $\delta^{18}O$ and ±0.02‰ (±1$\sigma$ SD; $n = 116$) for $\delta^{13}C$ verified against NBS 19. Fifty samples or ~15% of the entire time series were analysed in duplicate and triplicate with robust average reproducibility better than ±0.05‰ (±1$\sigma$ SD) for $\delta^{18}O$ and ±0.02‰ (±1$\sigma$ SD) for $\delta^{13}C$, which verified the homogeneity of our measured samples (Supplementary Fig. 5).

**Coral boron chemistry preparation.** For $\delta^{11}B$ extraction and analysis, ~50–55 mg of coral powder was milled from each annual banding as previously described. $\delta^{11}B$ extraction from 50 ± 1 mg of coral powder per sample was slightly modified from ref. [62]. All powder samples were pre-treated with a cleaning process prior to dissolution and analytical measurements to ensure that seawater salt crystals did not contaminate the powder samples. Each individual annual coral powder sample was washed in 50 ml centrifuge tubes with 2 ml of 18.2 MΩ milli-Q water on a benchtop shaker for 10 min and centrifuged at 6000 rpm for 5 min to concentrate the washed powder at the bottom so the water can be carefully removed by pipet.

The pre-treated coral powder samples were then dissolved in 1.25 ml of 1N HNO$_3$ (Optima-grade Ultrapure). Each coral powder solution as well as the international coral standard JCp-1[63], North Atlantic Seawater Standard VI (NASS VI), and the boric acid reference standard (NBS SRM 951) were purified with 60–80 mg of the anion exchange resin Amberlite IRA 743 following the Batch Method of ref. [62]. The amount of resin added to each solution was sufficient for 100% extraction of the boron. Briefly, each individual batch of tubes (coral samples and standards solutions) were slowly neutralised to a pH of 7–9 with 1N NH$_4$OH over 4 h on a benchtop shaker with constant pH checks every 30 min. Once the pH remained stable between 8 and 9 pH units, the resin-added solutions were allowed to absorb the boron overnight. After overnight absorption, the individual tubes were rinsed five times with 18.2 MΩ milli-Q water on the benchtop shaker prior to boron extraction from the resin. The concentrated boron was eluted from the resin using four successive volumes of 1.25 ml 0.1N HNO$_3$ (Optima-grade Ultrapure) for a total extracted volume of 5 ml. Each individual extraction steps (elution of 1.25

ml 0.1N HNO$_3$) lasted at least 4 h to maximise boron extraction yield. Finally, the boron concentrations were adjusted to 150–200 ppb in 0.1N HNO$_3$ (Optima-grade Ultrapure) for $\delta^{11}$B ratio analysis on the MC-ICP-MS.

**Coral boron isotope analysis**. Boron isotope composition ($\delta^{11}$B) was determined using a double-focusing sector-field Multi-Collector Inductively Coupled Plasma Mass Spectrometer (Thermo Neptune Plus MC-ICP-MS) on the Analytical Platform PANOPLY (LSCE-GEOPS), at LSCE, Gif-sur-Yvette, France following established methods[62]. The $\delta^{11}$B ratio MC-ICP-MS analysis results are reported in ‰ deviation relative to the NBS SRM 951 standard (boric acid isotopic standard). The standards from each individual batch (JCp-1, NBS SRM 951 and NASS VI) were first measured using the 'classic' method in the following sequence per standard: (a) rinse solution (0.1N HNO$_3$ and 0.04N HF), (b) blank solution (0.1N HNO$_3$, same solution as the boron extraction and final dilution): typical blank contribution is <~0.5%, and (c) the standard (JCp-1 or NBS SRM 951 or NASS VI). This process sequence is repeated until all the standards from each individual batch or batches are analysed. Upon the completion of this 'classic' method, a 'rapid' method sequence is initiated following the common standard-sample bracketing.

The standard-sample bracketing analytical method is used to systematically control the $\delta^{11}$B ratio mass drift over time. All the individual coral samples were prepared and measured in random sequences to exclude possible effects of chemistry by batch or analytical bias linked to MC-ICP-MS sequences. Each individual solution was measured in triplicate that included triplicate measurements per run per analysis. Thus, each solution was analysed nine times achieving excellent reproducibility and stability. The analytical uncertainty of $\delta^{11}$B ratio measurements for JCp-1 was robust ($24.28 \pm 0.15$‰, $n = 22$, $2\sigma$ SD; Supplementary Fig. 8) and within the previously published $2\sigma$ SD range for MC-ICP-MS from ref. [64].

Studies using *D. heliopora* coral at sub-seasonal resolutions have shown offsets between the different skeletal materials (columellar and septal) for sub-seasonally resolved isotope-based climate reconstructions[30–32]. Consequently, three sections of 20-year periods (1935–1954, 1856–1875 and 1768–1787) were removed from the same coral core slabs for replication and test of reproducibility. The powder samples were collected on the same coral slabs from a different coral polyp on the same growth banding, adjacent but not overlapping to the original sampling transect (Supplementary Fig. 2). Powders were processed under the same conditions to examine intra-colony reproducibility of the $\delta^{11}$B results.

**Coral-based seawater pH calculations**. Studies have shown that coral $\delta^{11}$B values record the pH of the extracellular calcifying fluid or space between the skeleton and the calicoblastic ectoderm and indirectly the ambient seawater pH[39,65]. Thus, the estimation of coral calcifying pH values (pH$_C$) were calculated from coral skeletal $\delta^{11}$B ratios following the established equation, Eq. (1) of ref. [34]:

$$pH_C = pK_B - \log\left[\left(\delta^{11}B_{SW} - \delta^{11}B_C\right) / \left\{\alpha_{[B3-B4]}\delta^{11}B_C - \delta^{11}B_{SW} + 10^3\left(\alpha_{[B3-B4]} - 1\right)\right\}\right] \tag{1}$$

The parameter pK$_B$[37,66] is the equilibrium (dissociation) constant for boron in seawater. For this study, we used the constant value, 8.58, because of the limited SST range (24–25 °C; Supplementary Fig. 6) and a stable annual sea surface salinity (SSS) of ~35 S$_p$. The value was not adjusted to ambient temperature and salinity because we do not know the true historical SST and SSS values beyond the instrumental records. Furthermore, studies have shown that large significant adjustments of SSS and SST based on monthly events such as floods have only a minimal impact on estimated pH (<0.01 pH units) after correction of pK$_B$[16]. In the above equation, $\delta^{11}B_{SW}$ (39.61‰[67]) and $\delta^{11}B_C$ represent the $\delta^{11}$B in seawater and coral, respectively. The $\alpha_{[B3-B4]}$ is the fractionation factor for isotope exchange between boric acid [B(OH)$_3$] and borate [B(OH)$_4^-$] in seawater. For this reconstruction, we applied the commonly used theoretically calculated fractionation factor ($\alpha_{[B3-B4]}$) of 1.0272[35] to facilitate comparison to previously published studies of $\delta^{11}$B pH reconstructions[12–17] and is also consistent with observations. The estimated seawater pH (pH$_{sw}$) (Fig. 2a) value was then converted from the above calculated pH$_C$ using the following equation, Eq. (2), based on *Porites* spp. from ref. [39]:

$$pH_{sw} = (pH_C - 5.954)/0.32 \tag{2}$$

**Statistical analyses**. We used the Pearson's product-moment correlation analysis to test the relationship between the different proxies ($\delta^{11}$B, $\delta^{13}$C and $\delta^{18}$O) of *D. heliopora* (Supplementary Fig. 9). Time series analyses on the proxy records were completed using the multi-taper method (MTM) and singular spectrum analysis (SSA) in kSpectra Toolkit[68]. SSA was applied as a nonparametric estimation technique based on principal component analysis to decompose time series into several significant frequency components allowing for the quantification of variance[69]. The principal or most dominant variance found by SSA in the coral proxy records is the long-term secular trend (Fig. 2), which was removed (detrended) for subsequent analyses. To estimate the power spectral density and

significance of each record, MTM spectral analysis[69] was completed on the detrended proxy records (Supplementary Fig. 11). Wavelet and cross-wavelet coherence analysis[70,71] (Fig. 7 and Supplementary Fig. 12) was also completed to examine the relationships and coherences between the proxy records and assessed by Monte Carlo methods ($\rho = 0.05$)[71] for probability distribution.

Change point analysis was performed on the individual *D. heliopora* isotope records using the Bayesian change point algorithm[72] in MATLAB to determine the exact timing of regime boundaries or significant changes in each proxy's secular trend. The change point analysis allows for the assessment of the median initiation points or the onset of changes in the trend and provides estimates of uncertainty and confidence intervals. Since the time series are in violation of the independent error structure (serial correlation), each coral proxy time series were time averaged over 8–10-year windows prior to running the change point analysis algorithm.

**Data availability**. The data reported in this paper have been deposited and fully available without restrictions at the information system PANGAEA (Data Publisher for Earth and Environmental Science; https://doi.pangaea.de/10.1594/PANGAEA.886966).

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

## Acknowledgements

Primary funding for this research was supported by a fellowship to H.C.W. with funding to D.D. and A.T. from the Institute Pierre Simon Laplace (Labex L-IPSL) under the Work Packages 4 and 5, IMPACTS: Project 16 – Impacts of climate change (ocean acidification and warming) on corals, which is provided by The French National Research Agency, ANR (Grant no: ANR-10-LABX-0018). Additional funding support was provided by The

French National Research Agency, ANR Project CARBORIC (Grant no: ANR-13-BS06-0013-04) to E.D., D.B. and by an internal research project support funded by the IPSL to C.E.L., D.D. and E.D. We thank the following people for their assistance: John Butscher of the IRD-Centre de Noumea, New Caledonia for field campaign coordination and assistance with coral core collection; Sandrine Caquineau of the IRD-France Nord for SEM and coral powder XRD analyses support, these instruments are a part of the analytical platform ALYSES (IRD-UPMC) funded by grants from Région lIe-de-France; Dr. Stéphane Lenoir of the Department of Radiology at the L'Institut Mutualiste Montsouris Paris for coral scanning assistance; Fatima Manssouri of the LSCE for stable isotope analysis assistance. This is LSCE contribution number 6452.

## Author contributions

D.D. and A.T. initiated the original project in collaboration with E.D., D.B., C.E.L. and F.L.C. D.D. and H.C.W. designed the research. H.C.W. completed the coral sampling, sample processing, data analysis and served as primary author on this manuscript. C.E.L. through a collaborative project with E.D., D.D. and F.L.C. recovered the coral core. C.E.L. assisted H.C.W. in SEM and XRD analyses. L.B. and H.C.W. performed the $\delta^{11}$B extraction chemistry. L.B., H.C.W., A.D. and E.D. executed the $\delta^{11}$B ratio analysis. D.B. and H.C.W. completed the $\delta^{13}$C and $\delta^{18}$O ratios analyses. E.P.B. completed the $^{230}$Th/U-age determination. All of the authors assisted in interpretation, editing, discussed the results and wrote the manuscript.

## Additional information

**Competing interests:** The authors declare no competing interests.

