## [Peer Review File · Nature Communications]

Reviewers' comments:

Reviewer #1 (Remarks to the Author):

The authors claim to have an excellent record of surface ocean pH, SST and CO₂ uptake in the southwest Pacific. The analysis is based on state of the art geochemistry of coral cores. Three proxies were used that track surface ocean properties or internal pH of the calcifying fluid in coral skeletons. The analysis was done at annual resolution for a period of 323 years on a slow-growing coral *Diploastrea*.

The data are very well presented and the paper is written in a logic way. Uncertainties of measurements are reported and appropriate tests for reproducibility of the Boron isotope proxy were added. Comparisons to other studies from the Pacific were added that help to constrain the finding of the present study.

To me the data and their interpretation are very convincing and of high interest to a broad readership. Records of ocean acidification are still scarce and this record adds an important dataset from a very unique ocean setting. The Supplement provides very important additional information that helps the reader to grasp the ideas of the manuscript.

I have a few minor suggestions that might improve the value of the paper. First of all, the authors show a comparison of the Nino3.4 index with the coral Boron record. What I would like to see more in detail is the actual signatures during major El Nino and La Nina years. What is the direction of change in Boron during a strong El Nino or La Nina? Is there a consistent relationship and does it follow the expected pH change during a El Nino or La Nina at New Caledonia? I can see a few prominent years or series of years where Boron shows a strong shift to more positive or negative values, does this coincide with a particular phase of ENSO? For instance 1790's and early 1800's, between 1890 and 1920, the latter a period of strong ENSO variability.

I also see anti-phasing between $\delta^{18}O$ and Boron, is it an indication for a link between mean SST and pH on interannual and decadal cycles?

The last remark is on the secular trend in Boron. There is a clear shift in Boron after the 1890's to lower values with almost 100 years before at high mean values, and again low Boron before 1800. While the recent shift towards the 20th century can be related to anthropogenic ocean acidification, the early period pre-1800 of low Boron or pH can't be. What could be the reason for the low Boron pre-1800? Is it the cooler mean SST at that time which led to naturally higher sink for CO₂ in the SW Pacific? Normally colder water is more susceptible to CO₂ uptake. Can the authors provide a mechanism or confirm SST as a potential link?

A last question is regarding the conversion of internal pH to seawater pH since I am not an expert in Boron isotope chemistry. Is the conversion chosen by the authors an accepted way of relating internal pH to seawater pH? I ask because we know by now that Boron gives us a valuable record of internal pH regulation by the coral. It means that the Boron helps us to decipher the calcification process of the coral driven by SST. How well do we know the offset between internal pH and seawater pH? The authors mention that they wanted to focus on relative changes in pH, yet an absolute value reconstruction is provided in Figs 3 and 5. Is that generally accepted in the Boron proxy community?

Overall, I recommend publication after addressing the comments.

Review: Acidification of the south Pacific surface ocean from anthropogenic CO₂ uptake over the last 3 centuries
Wu et al.

The authors present $\delta^{11}\text{B}$, $\delta^{13}\text{C}$ and $\delta^{18}\text{O}$ data measured in coral core material representing 300 years of coral growth. These data are used to reconstruct respectively the pH, $\delta^{13}\text{C}$ of dissolved inorganic carbon and temperature of surface seawater at this South Pacific site. They find a decline in $\delta^{11}\text{B}$ (decrease in pH) since ~1880 that is accompanied by a decrease in $\delta^{13}\text{C}$. Covariance between $\delta^{11}\text{B}$ and $\delta^{13}\text{C}$ suggest a common causal link between the drop in pH and decrease in surface ocean DIC; the incursion of anthropogenic carbon. Variability in these isotopic records shows a coupling to reconstructed sea surface temperature ($\delta^{18}\text{O}$) and this variation occurs on a periodicity similar to that of the ENSO cycle. This suggests that changes in surface temperature and wind strength are key drivers of carbon uptake and seawater pH at this site.

I enjoyed reading this manuscript. Reconstructing surface water pH at high resolution, beyond current instrumental observations (going back only ~30yrs) is of clear importance to our understanding of the rates of ocean acidification and the impact on marine calcifiers. As such, the content should appeal to the wide readership of Nature Comms. I found many positives in the work that mean this has the potential to be a significant contribution to the field of coral geochemistry and ocean acidification research. These include:

- 1) The authors' choice of coral is a good one. *Diploastrea heliopora* grows at around half the rate of large Porites corals typically used for this purpose. This has allowed them to obtain an impressive 300-year sample section. Large polyps in this species are also beneficial during sampling.
- 2) The choice of a site that is currently a sink for atmospheric CO₂ is a sound choice when trying to reconstruct past changes in OA
- 3) As far as I am aware, before now this species of coral had not been measured for its boron isotopic composition. These new $\delta^{11}\text{B}$ data demonstrate that pH upregulation is occurring in this taxon similar to other shallow water corals and that it too confers a degree of resilience to changes in external pH
- 4) The samples have been thoroughly checked for their preservation giving confidence to the primary nature of $\delta^{11}\text{B}$ data.
- 5) The weight of $\delta^{11}\text{B}$ data and 300 year duration of the record are impressive, particularly in comparison to previous coral $\delta^{11}\text{B}$ records (Fig 5)
- 6) Replicate overlapping sections of down core $\delta^{11}\text{B}$ are a nice addition and show close agreement. Again, this gives confidence that trends are not an artefact of sampling and chemically heterogeneous parts of the coral have been avoided.
- 7) The link of coral $\delta^{11}\text{B}$ to ENSO pacing is compelling and an important finding.

Unfortunately, parts of the discussion were unclear and I found the reasoning a little stretched in places, therefore I cannot recommend the paper for publication in its current form. I do however strongly urge the authors make some changes to the

discussion and resubmit to Nature Comms. I do feel that these well-produced records are worthy of a place in a high impact journal for the reasons given above.

Joe Stewart

Points to address:

Coral d11B here is used as a proxy for seawater pH, however, as stated in the manuscript, coral d11B actually records the pH within the coral. While there are examples where this internal pH is strongly affected by external pH it is important to point out that seawater pH is being inferred by proxy. I think therefore the wording in places is perhaps a little too strong (e.g. Line 194). There have been many examples now where external pH is far from the only factor controlling internal pH (e.g. Comeau et al. 2017, McCulloch et al. 2016)

While I like the use of a new species for this d11B-pH proxy work. This does present some challenges when converting d11B to seawater pH as there is no direct calibration of this species. On line 196 it is stated that “conversion based on several aragonite coral genera” however it looks like only one coral species (*P. cylindrica*) has been chosen from the McCulloch paper. A better argument would be that this is a conservative choice from the McCulloch calibrations available; one that gives the minimum observed pH change because of the slope. Or that this one has been chosen as it is another massive coral that occupies a similar niche to your species here (i.e. it makes sense that you would avoid the branching acropora calibrations).

Calculations of pH from d11B data appear correct based on the assumptions made here, however without a data table listing the isotopic results it is difficult to properly check this working. A lot of work has clearly gone into generating these data therefore I strongly encourage the authors to show these d11B, d13C and d18O results in a supplementary table.

Some coherency is shown between d18O and temperature, but this relationship ($R^2=0.4$) is not that strong. The problem comes from calibrating across such a narrow range of temperatures (2 deg C). The $\delta^{18}\text{O}$ -SST sensitivity of -0.25 ‰ per degree C that is used however sounds sensible based on studies in other carbonates. The coral geochem community has been slowly moving away from this proxy for the salinity complications mentioned in the manuscript. If the authors have Sr/Ca or Li/Mg temperature data for this site I would encourage them to share it, however I understand that generating these data will take more work that is perhaps beyond the scope of this study focusing on OA.

I have some concerns over the interpretation of the pH trends. The authors are keen to highlight newsworthy declines in pH (e.g. decrease in d11B observed since 1880), but there is little acknowledgement or discussion of the causes of large pH rises that are also documented in the down core record. For example, the rise in d11B up until 1880 is as large in magnitude as the decline in pH towards the modern that has been dubbed OA. What might be the cause of this elevation in pH at this site? Most of the drop in

pH recorded in this core occurs in a large step around 1890. I notice from the core images that this is mighty close to a dark horizon in the core (just below the core break) that might represent a mortality event or an interval of reduced growth. It is difficult to see from the figures, but can the authors make a convincing case that this dark horizon is not influencing the d11B trends we see here? If it is a major stress event in the core, then the down core d11B could be interpreted as an increase up until 1880 then an interval of little pH change throughout the last century (see below)...

...While I think the authors are probably correct in their interpretation of the d11B trends. I think it is important to remove the possibility that this upper core section may represent a different phase of coral growth.

Similarly, the most recent 20 year decline in pH is noted (red arrow below), but this d11B fall is as large as the d11B rise from ~1975 to ~1990 (blue arrow). If OA is responsible for the downward trend over the last 20 years, what is responsible for the preceding rise of equally "striking" magnitude and duration? Might this just suggest that ENSO modulation is dominant driver on these timescales? I am not sure of the answer to this myself, particularly as the red arrow is in excellent agreement with the recorded pH decline at the Hawaii HOTs site.

More minor points:

Line58 with links

Line59 timescales

Line101 *Diploastrea heliopora* can be abbreviated to *D. heliopora* from here onwards (e.g. line112)

Line167 Suggest change "observe" to "reconstruct"

Line170 This reads as if the fractionation factor is pH dependent too. This needs rewording.

Line185: Suggest rewording here. Could perhaps be read as pH becoming higher. You mean a more consistent, homogeneous sampling protocol was achieved.

Line206 Delete “coefficient”

Line208 I do not follow the link here. This assumes that there should be a direct link between d11B and temperature. Strong relationships between d13C and d11B make sense as they are both measuring a component of the carbon cycle. However, I'm not sure such a direct link can be made between coral d11B and rates of ocean warming at this site. Both OA and SST warming are changing at different rates, with much spatial heterogeneity in the oceans. It seems strange to attack the fidelity of your ocean temperature record again when an a-priori link between these proxy records is unclear.

Line 219 section needs rewording. No need to say both "depleted in 13C" or "enriched in 12C"; one is implied by the other. Perhaps just state that The oceans are the major sink of anthropogenic CO2 and incursion of isotopically light carbon (sourced from burning of fossil fuels) into the oceans has caused a decrease in the d13C of DIC, termed the seuss effect.

Line 228 Delete “derived proxies”

Line228 Suggest “cannot be attributed to growth rate as vertical extension rates remained constant throughout the interval of study”

Try to avoid the word "since" when you mean "because", especially in this context as "since" has a dative connotation.

Line 237 Delete “comparison”

Line 239 Sounds as if the coral is acting as a CO2 sink for anthropogenic emissions. Rather this shows that the coral is recording d13C of DIC

Line 240 agreement with

Line 241 Can the porites record referred to in the text be made more visible in Fig 4?

Line 242 I'm not sure the d13C itself "confirms the modern OA crisis" (whereas the d11B record if controlled by external pH certainly does). All this d13C record confirms is that the source of carbon is likely atmospheric. On its own, it says nothing about the pH of the ocean or the response of marine calcifiers to the pH changes (i.e. the OA crisis)

Line 248 "...show pronounced low to high pH reversals and vice versa". Unclear what the authors are trying to say here. pH goes down and up, and up and down? Suggest rephrase

Line 256 This sentence is very unclear. It is strange to talk in orders of magnitude when pH is a log scale measurement. Do you mean simply the swings in pH are almost half that recorded over G-IG cycles?

Line 262 incredible is a strange choice of word. The pH fluctuations are large, but perfectly credible.

Line 278 I am not entirely convinced by the reasoning here. How is the veracity of this new coral pH record strengthened by it being at odds with existing geochemical coral data sets (albeit from restricted sites) and model data? I feel this sentence is a step too far.

Line 298 and 300 Delete. I think this is a given. Data were generated a world-leading boron isotope lab. One would not expect sample wash out to be a considerable influence and the reader would not expect the authors to be interpreting trends that are within analytical error.

Line 375 replace “due to” with “caused by”

Line 393 What does DPU stand for?

Line 457 Unless this in-house standard has been measured elsewhere and has certified values the averages obtained here mean little to the reader and give no further indication of accuracy. Suggest leaving these results as precision only.

Line 479 “in-house” repeated

Line 501 Too many details in this section. Uptake rate of the particular neb used for example is of no use to a reader looking to replicate these results in their lab. I presume most of these details (e.g. cones) are given in the established methods of ref 57 and can therefore be cut here.

Line 517 Was carry over a concern. How big were the blanks compared to samples? Was ammonia or HF used to help wash out?

Line 521 If JCt values are not quoted here they perhaps no need to mention them above. JCp is of course the more relevant standard here anyway.

Line 551 I fail to see the importance of using this fractionation factor calculation from the 1970s? Are the authors suggesting that the resultant low internal pH estimates are feasible? i.e. no pH upregulation at all.

Line 558 state here that this is the *Porites cylindrica* calibration from McCulloch

Fig2 Could Figs 2 and 3 not be combined? I feel that Fig 3C adds little. Couldn't this smoother just be added to Fig 3B

Fig4 Remove word “record” from line labels. Make sure law dome label is closer to the grey line; at the moment it is not clear which line it corresponds to. Change axis label to "Atmospheric d13C"

Fig5 Circle markers on the plots are misleading. Are they in place of a legend or do they bear some relation to the x and y axes on their respective plots? I presume not. Perhaps these should be moved next to the d11B labels

Fig7 caption states trend removed similar to fig 6. Should this be fig3?

Fig S6 line 89 typo in the word “year”

Response to reviewer comments

We sincerely thank all reviewers for the detailed and constructive comments that they have provided on our manuscript. These comments and suggestions have been very instructive and constructive in assisting us to improve the manuscript.

Below are our detailed responses to the reviewers' comments in 'blue' with the associated line numbers and text changes highlighted.

Reviewers' comments:

Reviewer #1 (Remarks to the Author):

The authors claim to have an excellent record of surface ocean pH, SST and CO₂ uptake in the southwest Pacific. The analysis is based on state of the art geochemistry of coral cores. Three proxies were used that track surface ocean properties or internal pH of the calcifying fluid in coral skeletons. The analysis was done at annual resolution for a period of 323 years on a slow-growing coral *Diploastrea*.

The data are very well presented and the paper is written in a logical way. Uncertainties of measurements are reported and appropriate tests for reproducibility of the Boron isotope proxy were added. Comparisons to other studies from the Pacific were added that help to constrain the finding of the present study.

To me the data and their interpretation are very convincing and of high interest to a broad readership. Records of ocean acidification are still scarce and this record adds an important dataset from a very unique ocean setting. The Supplement provides very important additional information that helps the reader to grasp the ideas of the manuscript.

I have a few minor suggestions that might improve the value of the paper. First of all, the authors show a comparison of the Nino3.4 index with the coral Boron record. What I would like to see more in detail is the actual signatures during major El Nino and La Nina years. What is the direction of change in Boron during a strong El Nino or La Nina?

We can identify distinct patterns of $\delta^{11}\text{B}$ or pH change during severe or very severe El

Niño (EN) and La Niña (LN) years. For this study, we used the Oceanic Niño Index (ONI) as the standard for the identification of ENSO events in the tropical Pacific. The events identification are calculated based on the Extended Reconstructed Sea Surface Temperature (ERSST) version 4 [Huang *et al.*, 2014], as the 3-month running mean SST anomaly that is above or below the 0.5°C threshold for a period of at least 5 months. The threshold is further broken down into categories of weak (± 0.5 -0.9°C), moderate (± 1.0 -1.4°C), severe (± 1.5 -2.0°C), and very severe ($> \pm 2.0$ °C). The ENSO events over the period 1950-2011 CE with the best instrumental SST record (ERSST v.4) for ONI ENSO identification are now listed in the new Table 2 to accompany the new Fig. 5.

During most EN events, the $\delta^{11}\text{B}$ signature decreases from the preceding year translating into a surface seawater decrease in pH, and the opposite occurs during LN events with an increase in $\delta^{11}\text{B}$ signature from the preceding year (increase in surface pH). The increase (decrease) in $\delta^{11}\text{B}$ -pH follows the expected patterns of warming (cooling) of SST and during periods of anomalous precipitation from ENSO events. One additional important observation is that not all severe or very severe EN events result in the same magnitude of $\delta^{11}\text{B}$ depletion. Some moderate EN events (based on ERSST amplitude) displayed more pronounced depletion of $\delta^{11}\text{B}$ than severe EN events, especially the very severe EN of 1997-98. Severe LN events are consistent with enrichment of $\delta^{11}\text{B}$ indicating an increase in pH.

To address this comment and suggestion, we have added a new Fig. 5 that highlights the behaviour of coral $\delta^{11}\text{B}$ (seawater pH) from 1950-2011 CE with the identified ENSO events listed in the new Table 2. We have also included additional explanation and observations in the main text on these recent ENSO events as explained in the previous paragraph. This new passage can now be found on lines 338-354: “During most El Niño (EN) events, the $\delta^{11}\text{B}$ signature decreases from the preceding year translating into a surface seawater decrease in pH, and the opposite occurs during La Niña (LN) events with an increase in $\delta^{11}\text{B}$ signature from the preceding year (increase in surface pH; Fig. 5; Table 2).

The most recent ENSO events indicate possible changes of up to $\pm 0.35\%$ in coral $\delta^{11}\text{B}$ (Fig. 5; Table 2), which can equate to ± 0.07 pH units change in seawater pH for a single event under the most severe conditions. An additional important

observation is that not all severe or very severe EN events result in the same magnitude of $\delta^{11}\text{B}$ depletion. Some moderate EN events (based on ERSST³³ amplitude; Oceanic Niño Index) displayed more pronounced depletion of $\delta^{11}\text{B}$ than severe EN events and the very severe EN of 1997-98. Severe LN events are consistent with enrichment of $\delta^{11}\text{B}$ indicating an increase in pH (Fig. 5; Table 2). In addition, periods of more 'active' ENSO activity⁵⁵ (1890s-1910s) are recorded in our coral as a series of major depletions indicating decreases in pH in New Caledonia. The fluctuations in our coral-based $\delta^{11}\text{B}$ -pH thus reflect the highly dependent nature of ocean CO_2 uptake across the air-sea surface interface following the pacing of ENSO.”

The new Figure 5 with caption and Table 2 in the revised manuscript are shown below in this Reply Comment.

Fig. 5. New Caledonia coral $\delta^{11}\text{B}$ and ENSO events.

The New Caledonia coral $\delta^{11}\text{B}$ record from 1950-2011 CE with recent ENSO events identified (Table 2). The severity of the listed El Niño (EN) or La Niña (LN) events are identified by the Oceanic Niño Index (ONI) from the 3-month running mean SST anomaly that is above or below the 0.5 °C threshold for a period of at least 5 months in the Niño 3.4 region based on ERSST version 4 (ref. ³³). The threshold of ENSO event is broken down into categories of moderate -M- (± 1.0 - 1.4 °C), severe -S- (± 1.5 - 2.0 °C), and very severe -VS- ($> \pm 2.0$ °C).

Table 2. ENSO Events from Oceanic Niño Index.

Historical ENSO events, El Niño (warm) and La Niña (cold), from the Oceanic Niño Index (ONI) calculated as the 3-month running mean SST anomaly in the Niño 3.4 region (ERSST ver. 4; ref. 33) that is above or below the 0.5°C threshold for a period of at least 5 months. The threshold is further broken down into categories of weak (± 0.5 - 0.9°C), moderate (± 1.0 - 1.4°C), severe (± 1.5 - 2.0°C), and very severe ($> \pm 2.0^\circ\text{C}$).

El Niño				La Niña			
Severe (S)	$\delta^{11}\text{B}$ change (‰)	Very Severe (VS)	$\delta^{11}\text{B}$ change (‰)	Moderate (M)	$\delta^{11}\text{B}$ change (‰)	Severe (S)	$\delta^{11}\text{B}$ change (‰)
1957-58	-0.079	1982-83	-0.323	1954-55	+0.056	1973-74	+0.235
1966-67	-0.187	1997-98	-0.029	1969-70	+0.357	1975-76	+0.259
1972-73	-0.115			1996-97	+0.054	1988-89	+0.128
1987-88	0.007					1998-99	+0.088
1992-93	-0.356					1999-00	+0.293
						2007-08	+0.184

Is there a consistent relationship and does it follow the expected pH change during a El Nino or La Nina at New Caledonia?

Yes, we do find a consistent relationship between the expected pH change and EN or LN. The $\delta^{11}\text{B}$ signature decreases translating into a decrease in pH from the preceding year during an EN. The opposite occurs during a LN with an increase in $\delta^{11}\text{B}$ signature (increase in pH). The increase (or decrease) in $\delta^{11}\text{B}$ -pH follow the expected patterns of warming (or cooling) of SST and during periods of anomalous precipitation during ENSO events.

I can see a few prominent years or series of years where Boron shows a strong shift to more positive or negative values, does this coincide with a particular phase of ENSO? For instance 1790's and early 1800's, between 1890 and 1920, the latter a period of strong ENSO variability.

We agree with this reviewer's assessment and based on historical ENSO events recorded in multiple proxy records [*Gergis and Fowler, 2009*], there are certainly more 'active' ENSO periods than others in the past. During time periods of higher ENSO activity (e.g. 1890s-1910s) with many high magnitude EN events recorded [*Gergis and Fowler, 2009*], our New Caledonia $\delta^{11}\text{B}$ record exhibits major depletions indicating decreases in pH at New Caledonia.

It is likely that these series of years with extended high magnitude EN events resulted in the shifts in $\delta^{11}\text{B}$ -pH that was pointed out by Reviewer 2 with the larger interannual oscillation of coral $\delta^{11}\text{B}$ -pH. The recent events are also coincident to the large-scale Interdecadal Pacific Oscillation (IPO) phase shifts that occurred in 1945, 1977, and 1999 [*Henley et al., 2015*]. The timing of these IPO shifts based on SST is coincident to our $\delta^{11}\text{B}$ -pH record. We have inserted additional statements in this revision to specifically indicate and better explain these shifts in our record. The sentence can now be found on lines 349-354: "Severe LN events are consistent with enrichment of $\delta^{11}\text{B}$ indicating an increase in pH (Fig. 5; Table 2). In addition, periods of more 'active' ENSO activity⁵⁵ (1890s-1910s) are recorded in our coral as a series of major depletions indicating decreases in pH in New Caledonia. The fluctuations in our coral-based $\delta^{11}\text{B}$ -pH thus reflect the highly dependent nature of ocean CO_2 uptake across the air-sea surface interface following the pacing of ENSO."

I also see anti-phasing between $\delta^{18}\text{O}$ and Boron, is it an indication for a link between mean SST and pH on interannual and decadal cycles?

Yes, there is definitely a direct link between the two isotope records ($\delta^{18}\text{O}$ and $\delta^{11}\text{B}$) where the colder water are more acidified (contain more CO_2) than warmer water and follows the principles of seawater carbonate chemistry in terms of temperature [*Zeebe and Wolf-Gladrow, 2001*]. Because of the partial temperature component in coral $\delta^{18}\text{O}$, there will definitely be a relationship to coral $\delta^{11}\text{B}$. Similar to our response to Reviewer 2, we have revised this description of the relationship between $\delta^{18}\text{O}$ and $\delta^{11}\text{B}$ in certain sections of the manuscript while also emphasizing the point that our coral $\delta^{18}\text{O}$ integrated both the temperature and the $\delta^{18}\text{O}$ of seawater or consequently the water balance at the sea surface. Thus, this anti-phase relationship between $\delta^{18}\text{O}$

and $\delta^{11}\text{B}$ may be observed on the interannual timescale but is rather ambiguous at the decadal-interdecadal timescale. The edited sentences can now be found on lines 331-332: “In general, coral $\delta^{11}\text{B}$ and $\delta^{18}\text{O}$ records display anti-phase coherency with higher SST concomitant with more acidic conditions.”

The last remark is on the secular trend in Boron. There is a clear shift in Boron after the 1890's to lower values with almost 100 years before at high mean values, and again low Boron before 1800. While the recent shift towards the 20th century can be related to anthropogenic ocean acidification, the early period pre-1800 of low Boron or pH can't be. What could be the reason for the low Boron pre-1800? Is it the cooler mean SST at that time which led to naturally higher sink for CO_2 in the SW Pacific? Normally colder water is more susceptible to CO_2 uptake. Can the authors provide a mechanism or confirm SST as a potential link?

Indeed, contrary to what is observed since the beginning of the Industrial Revolution, coral $\delta^{13}\text{C}$ does not follow the $\delta^{11}\text{B}$ signature in the oldest part of the record and remains stable while $\delta^{11}\text{B}$ shows acidic value. However, coral $\delta^{18}\text{O}$ does follow $\delta^{11}\text{B}$ coherently well with ‘colder’ concomitant values, which can partially explain the more acidic conditions as we have expressed in the previous reply comment. As we suggested in the manuscript describing the coral $\delta^{11}\text{B}$ -pH trend, the beginning of the $\delta^{11}\text{B}$ -pH record coincide with the emergence of the south Pacific from the Little Ice Age (Figs. 2 and 3) and possible oceanographic related changed. After which our coral $\delta^{11}\text{B}$ -pH is dominated by the effect of anthropogenic CO_2 as reflected in both $\delta^{13}\text{C}$ and $\delta^{11}\text{B}$. Our original wording attempted to reflect this possibility but was perhaps not clear enough for the reader. We have re-written the sentences accordingly on lines 314-327: “Before this onset of modern anthropogenic-driven OA, a coral $\delta^{11}\text{B}$ -pH maximum was reached in the late 1790s (Fig. 2). The increase in $\delta^{11}\text{B}$ at the centennial-scale from the 16th to the 17th century appears to be decoupled from coral $\delta^{13}\text{C}$ trend as the seawater DIC signature remained relatively level. We observed that the progressive increase in $\delta^{11}\text{B}$ -pH during this time period (1701-1761 CE) coincided with changes in temperature most likely linked to the termination of the LIA, a period that was documented to be 1.4 °C cooler at New Caledonia^{27,51}. The end of this period was however characterized by a maximum $\delta^{11}\text{B}$ -pH that is contrary to the recorded maximum $\delta^{18}\text{O}$ enrichment (cooler and/or more

saline conditions; Fig. 2). It is possible that as a consequence to the termination of the LIA, a redistribution of water masses occurred near New Caledonia, which experienced an intrusion of cooler and/or more saline water to the region from the enhanced subtropical countercurrent⁵² (Fig. S10). These connections point to the substantial linked behaviour of pH, temperature and salinity, at the longer-term centennial timescales.”

A last question is regarding the conversion of internal pH to seawater pH since I am not an expert in Boron isotope chemistry. Is the conversion chosen by the authors an accepted way of relating internal pH to seawater pH?

The conversion method used in this study is based on the seminal works of [Trotter *et al.*, 2011; McCulloch *et al.*, 2012a, 2017] and is currently the state-of-the-art method for the conversion of coral skeletal $\delta^{11}\text{B}$ signature to coral internal and seawater pH without a species-specific calibration. In the previously published coral $\delta^{11}\text{B}$ studies in Fig. 4, the results are derived from the same calibration methodology as our study if $\delta^{11}\text{B}$ is converted to pH. For this particular genus and species of coral, a pH reconstruction has never been published before.

I ask because we know by now that Boron gives us a valuable record of internal pH regulation by the coral. It means that the Boron helps us to decipher the calcification process of the coral driven by SST. How well do we know the offset between internal pH and seawater pH?

The offset between coral internal calcifying and external seawater pH is highly linear as diagrammed in Figure 4 of [Trotter *et al.*, 2011] for both temperate shallow water and tropical corals. Based on these relationships and other state-of-the-art studies on $\delta^{11}\text{B}$ -pH for corals, we now know that scleractinian corals up-regulate their internal pH at the site of calcification by approximately one-half of the ambient seawater pH [McCulloch *et al.*, 2012a, 2012b]. We have inserted this point in our manuscript to emphasize this relationship of internal-external pH and also on our carefully chosen calibration and conversion factors. This revised paragraph can be found on lines 189-207: “Careful considerations were taken to convert our coral $\delta^{11}\text{B}$ signature to coral internal pH as studies indicated differences between empirically- and theoretically-

determined fractionation factors for isotope exchange between boric acid and borate in seawater³⁵. Furthermore, results based on a variety of coral species across temperate and tropical regions indicate consistent difference between coral internal calcification pH by approximately one-half of ambient seawater pH^{38,39}. However, recent results also indicated possible modifications of coral internal pH due to external factors and species-specific processes^{40,41}. Thus, we estimated the changes in seawater pH at New Caledonia using a moderate general calibration conversion³⁹ (Fig. 2). To be conservative with our conversion, a calibration based on a similar massive coral species (*Porites* spp.) to the one used in this study (*D. heliopora*) was chosen allowing for the establishment of biological and seawater pH linkage to climate-driven patterns. We focused our discussion on the relative pH changes instead of the absolute pH values because rigorous palaeo-pH reconstructions from coral $\delta^{11}\text{B}$ signature require species-specific calibrations with robust quantification of physiological process (vital effects). Nevertheless, independently from a possible offset in our reconstructed pH using this calibration conversion, our observed trend and interannual to decadal variability from the measurements along the core will remain.”

The authors mention that they wanted to focus on relative changes in pH, yet an absolute value reconstruction is provided in Figs 3 and 5. Is that generally accepted in the Boron proxy community?

For this study, as just described above and now in the revised text, we used the calibration made on other massive tropical coral species (*Porites* spp.) as there are currently no other data available for the particular species described in this manuscript. Future work should and will be focused on calibrating the $\delta^{11}\text{B}$ -pH of this particular coral species based on in situ calibration because no other non-branching coral $\delta^{11}\text{B}$ -pH calibration data are available. Approximation of reconstructed pH value has been accepted in literature and is the reason why pH values were given within the manuscript based on our calculations. The precise boron isotopic signatures are also given and made publicly available for future works to translate our results into more precise pH values when more calibrations become available. Moreover, absolute seawater pH values may only be acquired once the robust quantification of internal-external pH offsets caused by physiological processes (vital effects) and the

validated α [B3–B4] value for this coral genus is established.

Nevertheless, if an offset from our reconstructed pH versus the actual seawater surface pH may be expected, again the trend and interannual to decadal acidification variability observed along the core should in no way be affected. Therefore, we believe that the focus on relative change in pH values is more important than the absolute pH values in this study. This is also the reason why the Figures presented in our manuscript along with most interpretations and discussions of the results are based on the coral $\delta^{11}\text{B}$ dataset and we cautiously only discuss the changes in terms of relative pH and not absolute pH changes.

See the above reply comment for the revised sentences.

Overall, I recommend publication after addressing the comments.

We thank this Reviewer for the thoughtful comments and recommendations that improved our manuscript.

Reviewer #2 (Remarks to the Author):

The authors present $\delta^{11}\text{B}$, $\delta^{13}\text{C}$ and $\delta^{18}\text{O}$ data measured in coral core material representing 300 years of coral growth. These data are used to reconstruct respectively the pH, $\delta^{13}\text{C}$ of dissolved inorganic carbon and temperature of surface seawater at this South Pacific site. They find a decline in $\delta^{11}\text{B}$ (decrease in pH) since ~1880 that is accompanied by a decrease in $\delta^{13}\text{C}$. Covariance between $\delta^{11}\text{B}$ and $\delta^{13}\text{C}$ suggest a common causal link between the drop in pH and decrease in surface ocean DIC; the incursion of anthropogenic carbon. Variability in these isotopic records shows a coupling to reconstructed sea surface temperature ($\delta^{18}\text{O}$) and this variation occurs on a periodicity similar to that of the ENSO cycle. This suggests that changes in surface temperature and wind strength are key drivers of carbon uptake and seawater pH at this site.

I enjoyed reading this manuscript. Reconstructing surface water pH at high resolution, beyond current instrumental observations (going back only ~30yrs) is of clear importance to our understanding of the rates of ocean acidification and the impact on marine calcifiers. As such, the content should appeal to the wide readership of Nature Comms. I found many positives in the work that mean this has the potential to be a significant contribution to the field of coral geochemistry and ocean acidification research. These include:

- 1) The authors' choice of coral is a good one. *Diploastrea heliopora* grows at around half the rate of large *Porites* corals typically used for this purpose. This has allowed them to obtain an impressive 300-year sample section. Large polyps in this species are also beneficial during sampling.
- 2) The choice of a site that is currently a sink for atmospheric CO_2 is a sound choice when trying to reconstruct past changes in OA
- 3) As far as I am aware, before now this species of coral had not been measured for its boron isotopic composition. These new $\delta^{11}\text{B}$ data demonstrate that pH upregulation is occurring in this taxon similar to other shallow water corals and that it too confers a degree of resilience to changes in external pH

- 4) The samples have been thoroughly checked for their preservation giving confidence to the primary nature of d11B data.
- 5) The weight of d11B data and 300 year duration of the record are impressive, particularly in comparison to previous coral d11B records (Fig 5)
- 6) Replicate overlapping sections of downcore d11B are a nice addition and show close agreement. Again, this gives confidence that trends are not an artefact of sampling and chemically heterogeneous parts of the coral have been avoided.
- 7) The link of coral d11B to ENSO pacing is compelling and an important finding. Unfortunately, parts of the discussion were unclear and I found the reasoning a little stretched in places, therefore I cannot recommend the paper for publication in its current form. I do however strongly urge the authors make some changes to the discussion and resubmit to Nature Comms. I do feel that these well-produced records are worthy of a place in a high impact journal for the reasons given above.

Joe Stewart

Points to address:

Coral d11B here is used as a proxy for seawater pH, however, as stated in the manuscript, coral d11B actually records the pH within the coral. While there are examples where this internal pH is strongly affected by external pH it is important to point out that seawater pH is being inferred by proxy. I think therefore the wording in places is perhaps a little too strong (e.g. Line 194). There have been many examples now where external pH is far from the only factor controlling internal pH (e.g. Comeau et al. 2017, McCulloch et al. 2016).

Yes, we absolutely agree with this comment that some other factors can influence internal pH and some appear to be environmental (e.g. warming) or even some species-specific related effect as demonstrated in [Comeau et al., 2017; McCulloch et al., 2017]. It is also true that both reconstructed pH records are inferred from the same coral proxy (pH of calcifying space and pH of seawater). The reviewer is thus correct that there might be bias in the pH reconstruction at each step: first when we

reconstruct the internal pH at the coral calcifying space and second when we add an additional source of bias when reconstructing seawater pH. However, in our case where we reconstruct pH along the same coral core, the trend of the reconstructed internal pH values within our core can be considered as reliable as whatever bias applies to the absolute pH value, the same bias is applied along the whole core and does not interfere with the reconstructed pH trend. Nevertheless, it must be said that any use of climate proxy, and especially as they have a biological component, we must "convert" the geochemical data into physical quantity with great humility. We have tempered down these emphasized points in the revised manuscript and especially this paragraph that can be found on lines 189-207.

While I like the use of a new species for this $\delta^{11}\text{B}$ -pH proxy work. This does present some challenges when converting $\delta^{11}\text{B}$ to seawater pH as there is no direct calibration of this species. On line 196 it is stated that "conversion based on several aragonite coral genera" however it looks like only one coral species (*P. cylindrica*) has been chosen from the McCulloch paper. A better argument would be that this is a conservative choice from the McCulloch calibrations available; one that gives the minimum observed pH change because of the slope. Or that this one has been chosen as it is another massive coral that occupies a similar niche to your species here (i.e. it makes sense that you would avoid the branching *Acropora* calibrations).

We agree with this reviewer's argument that the coral $\delta^{11}\text{B}$ to seawater pH conversion should be based on a coral species that is as similar as possible to our coral because of the lack of direct calibration. Thus the calibration made on other massive tropical coral species (*Porites* spp.) was chosen [McCulloch et al., 2012a] as no data are available so far for *Diploastrea heliopora*. Future work should focus on calibrating the $\delta^{11}\text{B}$ -pH of this particular coral species based on in situ calibration. The revised sentences can now be found on lines 196-204: "Thus, we estimated the changes in seawater pH at New Caledonia using a moderate general calibration conversion³⁹ (Fig. 2). To be conservative with our conversion, a calibration based on a similar massive coral species (*Porites* spp.) to the one used in this study (*D. heliopora*) was chosen allowing for the establishment of biological and seawater pH linkage to climate-driven patterns. We focused our discussion on the relative pH changes instead of the absolute pH values because rigorous palaeo-pH reconstructions from coral $\delta^{11}\text{B}$

signature require species-specific calibrations with robust quantification of physiological process (vital effects).”

Calculations of pH from d11B data appear correct based on the assumptions made here, however without a data table listing the isotopic results it is difficult to properly check this working. A lot of work has clearly gone into generating these data therefore I strongly encourage the authors to show these d11B, d13C and d18O results in a supplementary table.

We are in full agreement with the new transparency standards of scientific publications in making all research data available. In fulfilling this requirement, we have uploaded our complete dataset presented in this study to PANGAEA, the Data Publisher for Earth & Environmental Science (<https://pangaea.de>). We have added an additional Heading with declaration in the manuscript at the end of the Methods section: “**Data availability**

The New Caledonia coral data reported in this paper has been deposited at the information system PANGAEA (Data Publisher for Earth and Environmental Science; <https://doi.pangaea.de/10.1594/PANGAEA.886966>).”

In this way, relevant information for this dataset in terms of collection site, collection method, sample processing, analyses, and etc. can all be easily accessed and processed.

Some coherency is shown between $\delta^{18}\text{O}$ and temperature, but this relationship ($R^2=0.4$) is not that strong. The problem comes from calibrating across such a narrow range of temperatures (2 deg C). The $\delta^{18}\text{O}$ -SST sensitivity of -0.25 ‰ per degree C that is used however sounds sensible based on studies in other carbonates. The coral geochem community has been slowly moving away from this proxy for the salinity complications mentioned in the manuscript. If the authors have Sr/Ca or Li/Mg temperature data for this site I would encourage them to share it, however I understand that generating these data will take more work that is perhaps beyond the scope of this study focusing on OA.

As this reviewer pointed out, the relationship between $\delta^{18}\text{O}$ and SST is not very obvious and is most likely influenced by salinity and/or precipitation. We have

revised and extended this point in the revised manuscript on lines 127-130: “Previous studies from the Pacific have demonstrated that coral $\delta^{18}\text{O}$ variations over multiple timescales represent changes of SST, SSS, precipitation, or a mixture of those parameters depending on the environmental setting^{18,27,28}..” And on lines 213-215: “Moreover, the correlation between $\delta^{11}\text{B}$ and $\delta^{18}\text{O}$ is also significant ($R = 0.42$, $p < 0.01$, $n = 319$; Fig. S9b) but indicates a weaker linear dependence because coral $\delta^{18}\text{O}$ record integrates both SST and the $\delta^{18}\text{O}_{\text{sw}}$ (water balance at the sea surface).”

However, we do not believe that the Sr/Ca or Li/Mg data will provide anything more convincing. With the Li/Mg thermometer, we will find the opposite problem to that observed at low temperatures: that is to say a strong annual variability of SST (2-3 °C) for the Li/Mg ratio (1.5/1.7 mmol/mol) is relatively constant. Thus, we are uncertain if the additional trace element records bring relevant information to the manuscript, as it will also contain inherent issues of proxy reconstructions. In addition, precise genus-specific coral-based Li/Mg–SST calibration at the annual or seasonal resolutions for *D. heliopora* is not currently available and such genus is not considered in the published calibration curves proposed in [Hathorne et al., 2013; Montagna et al., 2014].

Instead, we chose to focus this manuscript on the isotopic data, as these results are the most relevant to the subject matter considered here and will not be distracted by additional proxies and their issues. Although we understand the remark of the reviewer pointing out that the coral palaeoclimate community has been moving away from the interpretation of the $\delta^{18}\text{O}$ signature as an absolute SST proxy. There are however many robust basin-scale SST compilations derived from coral $\delta^{18}\text{O}$ signature, which still remains a relevant SST proxy [e.g. Tierney et al., 2015].

I have some concerns over the interpretation of the pH trends. The authors are keen to highlight newsworthy declines in pH (e.g. decrease in d11B observed since 1880), but there is little acknowledgement or discussion of the causes of large pH rises that are also documented in the down core record. For example, the rise in d11B up until 1880 is as large in magnitude as the decline in pH towards the modern that has been dubbed OA.

What might be the cause of this elevation in pH at this site?

We agree with this reviewer that there are other time periods of great interest recorded in our $\delta^{11}\text{B}$ signature. To be forthright with the readership, we have chosen to tone down our interpretation, which starts with a revised manuscript title. Instead of pointing out only the current OA period, our title now reflects the entire reconstruction, “**Surface ocean pH variations since 1689 CE and recent ocean acidification in the tropical south Pacific.**”

Similar to our response to Reviewer 1: Indeed, contrary to what is observed since the beginning of the Industrial Revolution, coral $\delta^{13}\text{C}$ does not follow the $\delta^{11}\text{B}$ signature in the oldest part of the record and remains stable because of the lack of Suess Effect while $\delta^{11}\text{B}$ shows acidic value. However, coral $\delta^{18}\text{O}$ does follow $\delta^{11}\text{B}$ coherently well with ‘colder’ concomitant values, which can partially explain the more acidic conditions as we have expressed in the previous reply comment. As we suggested in the manuscript on lines 314-327, describing the trend of the earlier part of the coral $\delta^{11}\text{B}$ -pH record, the changes in $\delta^{11}\text{B}$ -pH coincide with the Little Ice Age (Figs. 2 and 3) up to ~1800s and perhaps the intrusion of cooler and/or more saline water to the region based on our $\delta^{18}\text{O}$ record. After the 19th century, our coral $\delta^{11}\text{B}$ -pH is most likely dominated by the effect of anthropogenic CO_2 as reflected in the coherent $\delta^{13}\text{C}$ and $\delta^{11}\text{B}$ signatures. Our original wording attempted to reflect this possibility but was perhaps not clear enough for the reader. We have re-written the sentences accordingly: “**Before this onset of modern anthropogenic-driven OA, a coral $\delta^{11}\text{B}$ -pH maximum was reached in the late 1790s (Fig. 2). The increase in $\delta^{11}\text{B}$ at the centennial-scale from the 16th to the 17th century appears to be decoupled from coral $\delta^{13}\text{C}$ trend as the seawater DIC signature remained relatively level. We observed that the progressive increase in $\delta^{11}\text{B}$ -pH during this time period (1701-1761 CE) coincided with changes in temperature most likely linked to the termination of the LIA, a period that was documented to be 1.4 °C cooler at New Caledonia^{27,51}. The end of this period was however characterized by a maximum $\delta^{11}\text{B}$ -pH that is contrary to the recorded maximum $\delta^{18}\text{O}$ enrichment (cooler and/or more saline conditions; Fig. 2). It is possible that as a consequence to the termination of the LIA, a redistribution of water masses occurred near New Caledonia, which experienced an intrusion of cooler and/or more saline water to the region from the enhanced subtropical countercurrent⁵² (Fig.**

S10). These connections point to the substantial linked behaviour of pH, temperature and salinity, at the longer-term centennial timescales.”

Most of the drop in pH recorded in this core occurs in a large step around 1890. I notice from the core images that this is mighty close to a dark horizon in the core (just below the core break) that might represent a mortality event or an interval of reduced growth. It is difficult to see from the figures, but can the authors make a convincing case that this dark horizon is not influencing the d11B trends we see here?

We agree with this reviewer that there appears to be ‘darker’ banding in certain areas of the coral core but these areas are not particularly severe as the ones witnessed in massive coral mortality events (e.g. 1997-1998 El Niño [*Cantin and Lough, 2014*]). This particular ‘darker’ area (depth horizon) was also the region where skeletal SEM and microstructure analyses were conducted adjacent. These samples were not found to be anomalous when compared to the samples from the older section of coral core. Moreover, this particular ‘darker’ banding occurred after the 1900 CE growth year and was not during the period in question ~1890. Most importantly we do not observe a reduction in linear extension or a growth hiatus years within our core after these darkened areas, which would likely be observed if a major stress event had occurred.

If it is a major stress event in the core, then the down core d11B could be interpreted as an increase up until 1880 then an interval of little pH change throughout the last century (see below)...

...While I think the authors are probably correct in their interpretation of the d11B

trends. I think it is important to remove the possibility that this upper core section may represent a different phase of coral growth.

Possible stress event such as bleaching may lead to a depletion in coral $\delta^{11}\text{B}$ [Dishon et al., 2015] but these stress events remain inconclusive as other experiments revealed no bleaching impact on coral $\delta^{11}\text{B}$ [Schoepf et al., 2014]. The bleaching analysis of [Dishon et al., 2015] revealed a drop of -5.1‰ in coral $\delta^{11}\text{B}$, which is far greater in magnitude than the total analytical range of this study. We agree with this reviewer that a large ‘shift’ occurred around 1890, however, this change is nearly the same as the change in 1800. The plotting of these 2 simple linear trend lines is somewhat misleading as it does not take into account the nuanced long term secular trends change as indicated in our revised Fig. 2 and original Fig. 3 (now removed). In our previous responses, we postulate that these large changes likely occurred during periods of more ‘active’ ENSO or during time periods of higher ENSO activity (e.g. 1890s-1910s) with many high magnitude EN events recorded [Gergis and Fowler, 2009]. It is probable that these series of years with extended high magnitude EN events resulted in these larger interannual oscillation of coral $\delta^{11}\text{B}$ -pH. Finally, the recent ‘shifts’ are also coincident to the large-scale Interdecadal Pacific Oscillation (IPO) phase shifts that occurred in 1945, 1977, and 1999 [Henley et al., 2015] and supports our argument that it is not due to growth related bias.

Similarly, the most recent 20 year decline in pH is noted (red arrow below), but this $\delta^{11}\text{B}$ fall is as large as the $\delta^{11}\text{B}$ rise from ~1975 to ~1990 (blue arrow). If OA is responsible for the downward trend over the last 20 years, what is responsible for the preceding rise of equally "striking" magnitude and duration?

Please refer to our response below the next comment.

Might this just suggest that ENSO modulation is dominant driver on these timescales? I am not sure of the answer to this myself, particularly as the red arrow is in excellent agreement with the recorded pH decline at the Hawaii HOTs site.

During time periods of higher ENSO activity (e.g. 1890s-1910s) with many recorded high magnitude EN events [Gergis and Fowler, 2009], our New Caledonia $\delta^{11}\text{B}$ record exhibit major depletions indicating decreases in pH at New Caledonia. The recent events as pointed by this reviewer are also coincident to the large-scale Interdecadal Pacific Oscillation (IPO) phase shifts that occurred in 1945, 1977, and 1999 [Henley et al., 2015]. The timing of these IPO phase shifts based on SST is coincident to our $\delta^{11}\text{B}$ -pH record. It is likely that these series of years are also years with extended high magnitude EN events and resulted in the large interannual oscillation of coral $\delta^{11}\text{B}$ -pH. We have added additional sentences in the revised manuscript communicating these coherences. The new sentences are found on lines 338-354: “During most El Niño (EN) events, the $\delta^{11}\text{B}$ signature decreases from the preceding year translating into a surface seawater decrease in pH, and the opposite occurs during La Niña (LN) events with an increase in $\delta^{11}\text{B}$ signature from the preceding year (increase in surface pH; Fig. 5; Table 2)

The most recent ENSO events indicate possible changes of up to $\pm 0.35\%$ in coral $\delta^{11}\text{B}$ (Fig. 5; Table 2), which can equate to ± 0.07 pH units change in seawater pH for a single event under the most severe conditions. An additional important observation is that not all severe or very severe EN events result in the same magnitude of $\delta^{11}\text{B}$ depletion. Some moderate EN events (based on ERSST ³³ amplitude; Oceanic Niño Index) displayed more pronounced depletion of $\delta^{11}\text{B}$ than severe EN events and the very severe EN of 1997-98. Severe LN events are consistent with enrichment of $\delta^{11}\text{B}$ indicating an increase in pH (Fig. 5; Table 2). In addition, periods of more ‘active’ ENSO activity ³⁵ (1890s-1910s) are recorded in our coral as a series of major depletions indicating decreases in pH in New Caledonia. The

fluctuations in our coral-based $\delta^{11}\text{B}$ -pH thus reflect the highly dependent nature of ocean CO_2 uptake across the air-sea surface interface following the pacing of ENSO.”

More minor points:

Line 58 with links

Revised.

Line 59 timescales

Revised.

Line 101 *Diploastrea heliopora* can be abbreviated to *D. heliopora* from here onwards (e.g. line 112)

The coral genus is now abbreviated after the first mention in the main text.

Line 167 Suggest change “observe” to “reconstruct”

The word “observe” has now been removed and is replaced with “reconstruct”.

Line 170 This reads as if the fractionation factor is pH dependent too. This needs rewording.

Yes, this reviewer is absolutely correct in that the fractionation factor at equilibrium ($\alpha[\text{B}_3\text{-B}_4]$, see [Klochko *et al.*, 2006]) is independent of the pH by definition. We have revised the awkward wording in this sentence. This revised sentence can now be found on lines 169-172: “In seawater, the relative abundance of the two aqueous boron species (boric acid and borate) as well as their isotopic composition are pH dependent³⁴ with a constant fractionation factor between the two aqueous boron species³⁵”

Line 185: Suggest rewording here. Could perhaps be read as pH becoming higher. You mean a more consistent, homogeneous sampling protocol was achieved.

We agree with the reviewer that the awkward wording in this sentence may be misinterpreted or misunderstood. We have revised this entire sentence and it can now be found on lines 182-185: “The large amount of coral skeletal material used for $\delta^{11}\text{B}$ analysis (~50 mg) compared to traditional $\delta^{18}\text{O}$ and $\delta^{13}\text{C}$ analyses makes it statistically likely that the sample comprises all different skeletal structures (columellar and septal) and is thus representative of the whole coral skeleton.”

Line 206 Delete “coefficient”

Removed from revised manuscript.

Line 208 I do not follow the link here. This assumes that there should be a direct link between $\delta^{11}\text{B}$ and temperature. Strong relationships between $\delta^{13}\text{C}$ and $\delta^{11}\text{B}$ make sense as they are both measuring a component of the carbon cycle. However, I'm not sure such a direct link can be made between coral $\delta^{11}\text{B}$ and rates of ocean warming at this site. Both OA and SST warming are changing at different rates, with much spatial heterogeneity in the oceans. It seems strange to attack the fidelity of your ocean temperature record again when an a-priori link between these proxy records is unclear.

We want to point out that there is a link between the two isotope records ($\delta^{18}\text{O}$ and $\delta^{11}\text{B}$) where colder waters are more acidified (contain more CO_2) than warmer waters [Zeebe and Wolf-Gladrow, 2001]. We welcome this critical comment on our manuscript and we have now revised the description of the relationship between $\delta^{18}\text{O}$ and $\delta^{11}\text{B}$ placing it in more positive form indicating that $\delta^{18}\text{O}$ integrates both the temperature and the $\delta^{18}\text{O}$ of seawater, or consequently the water balance at sea surface. The revised sentence can now be found on lines 213-215: “Moreover, the correlation between $\delta^{11}\text{B}$ and $\delta^{18}\text{O}$ is also significant ($R = 0.42$, $p < 0.01$, $n = 319$; Fig. S9b) but indicates a weaker linear dependence because coral $\delta^{18}\text{O}$ record integrates both SST and the $\delta^{18}\text{O}_{\text{sw}}$ (water balance at the sea surface).”

Line 219 section needs rewording. No need to say both "depleted in ^{13}C " or "enriched in ^{12}C "; one is implied by the other. Perhaps just state that The oceans are

the major sink of anthropogenic CO₂ and incursion of isotopically light carbon (sourced from burning of fossil fuels) into the oceans has caused a decrease in the δ¹³C of DIC, termed the seuss effect.

We have revised the original two sentences in the Discussion describing the ¹³C Suess effect based on this reviewer comment. The new sentence can now be found on lines 221-224: “As the ocean is one of the major global sinks of anthropogenic CO₂ emission, the incursion of isotopically light carbon (¹²C) from the burning of fossil fuel into the ocean has caused considerable decrease in the δ¹³C of seawater DIC, known as the ¹³C Suess effect.”

Line 228 Delete “derived proxies”

Removed from revised manuscript.

Line 228 Suggest “cannot be attributed to growth rate as vertical extension rates remained constant throughout the interval of study” Try to avoid the word "since" when you mean "because", especially in this context as "since" has a dative connotation.

Based on this reviewer suggestion, we edited the sentences to state that the δ¹¹B changes are not related to the linear vertical extension growth with careful selection of our vocabulary to avoid any confusion. The new sentences can now be found on lines 230-233: “The geochemical variations observed in our coral cannot be attributed to the linear growth rate because the vertical extension rates remained relatively constant (2.68 ± 0.64 mm year⁻¹) throughout the interval of this study.”

Line 237 Delete “comparison”

Removed from revised manuscript.

Line 239 Sounds as if the coral is acting as a CO₂ sink for anthropogenic emissions. Rather this shows that the coral is recording δ¹³C of DIC

The original sentence has been revised for clarification based on this reviewer comment. The new sentence can now be found on lines 239-243: “These consistent secular trends from instrumental measurements and the rapid rate of coral $\delta^{13}\text{C}$ depletion found at New Caledonia (-0.024‰ yr^{-1} ; Table S3) recording the $\delta^{13}\text{C}$ of seawater DIC over the same period (1978-2011; Fig. 3 and Table S3) demonstrate the significant absorption of anthropogenic CO_2 emission by the oceans.”

Line 240 agreement with Line 241 Can the *Porites* record referred to in the text be made more visible in Fig 4?

The *Porites lutea* $\delta^{13}\text{C}$ record from New Caledonia [Quinn et al., 1998] in the revised new Fig. 3 of central and western Pacific coral records is now made more visible by making the line dashed and more bold. The legends in this figure have also been revised based on the Reviewer comments.

Line 242 I'm not sure the $\delta^{13}\text{C}$ itself "confirms the modern OA crisis" (whereas the $\delta^{11}\text{B}$ record if controlled by external pH certainly does). All this $\delta^{13}\text{C}$ record confirms is that the source of carbon is likely atmospheric. On its own, it says nothing about the pH of the ocean or the response of marine calcifiers to the pH changes (i.e. the OA crisis)

We have attempted our best to tone down the manuscript following this reviewer comment and have removed this phrase, “modern OA crisis” in this revised sentence. The new sentence can now be found on lines 243-247: “The pronounced long-term depletion since 1843 by at least 1.0‰ in our *D. heliopora* $\delta^{13}\text{C}$ record is in agreement to the nearest New Caledonia *Porites* sp. $\delta^{13}\text{C}$ record that found a depletion of 0.9‰ since the late 19th century²⁷ confirming the modern uptake of anthropogenic atmospheric CO_2 by the tropical oceans.”

Line 248 "...show pronounced low to high pH reversals and vice versa". Unclear what the authors are trying to say here. pH goes down and up, and up and down? Suggest rephrase

This sentence has been revised to clarify the statement on pronounced reconstructed

pH changes on both interannual to interdecadal timescales in our record. The new sentence can now be found on lines 251-253: “On both interannual and decadal-interdecadal timescales, the annually resolved $\delta^{11}\text{B}$ values as a proxy of pH show pronounced oscillations (Fig. 2)”

Line 256 This sentence is very unclear. It is strange to talk in orders of magnitude when pH is a log scale measurement. Do you mean simply the swings in pH are almost half that recorded over G-IG cycles?

We now recognize our mistake and following this reviewer comment; the sentence has been revised to clarify the statement on the magnitude of pH change recorded in our coral. The new sentence can now be found on lines 253-260: “These high-amplitude $\delta^{11}\text{B}$ or estimated pH variations in the surface waters in the South Pacific indicate large interannual changes of up to 0.8‰ equating to a 0.16 pH unit change in seawater (Fig. 2). The extremes of interdecadal fluctuations are smaller than those observed at the higher interannual frequencies with an average change of 0.4‰ in $\delta^{11}\text{B}$ or 0.08 pH unit in seawater. The swings between 0.08 to 0.16 pH units in South Pacific pH over interannual to interdecadal timescales are not approaching the high benchmark of 0.4 pH units for end-of-century emission scenarios as predicted by model simulations¹.”

Line 262 incredible is a strange choice of word. The pH fluctuations are large, but perfectly credible.

We now refer to the reconstructed pH fluctuations from the Great Barrier Reef [*Wei et al.*, 2009] and the South China Sea [*Wei et al.*, 2015] as “large” from this reviewer suggestion, which may be due to local reef effect or coastal riverine input. The new sentences can now be found on lines 264-270: “Moreover, studies from the South China Sea¹⁴ and the GBR¹⁵ recorded large interannual to interdecadal fluctuations (0.6-0.7 pH units) that are even greater than the 0.4 pH units predicted for the end-of-the-century (Fig. 4). The large pH discrepancy of the near-shore records from the South China Sea¹⁴ and GRB¹⁵ may indicate the large spatial variability of pH and possible coastal riverine influence¹⁶ or local reef effect as compared to our open-ocean record.”

Line 278 I am not entirely convinced by the reasoning here. How is the veracity of this new coral pH record strengthened by it being at odds with existing geochemical coral data sets (albeit from restricted sites) and model data? I feel this sentence is a step too far.

The original goal of this paragraph was to point out the possibilities and reasons for model simulation and proxy records discrepancy. This final sentence of the paragraph on magnitudes of pH change has been revised for clarification and toned down. The new sentence can now be found on lines 279-281: “Thus, our coral $\delta^{11}\text{B}$ record appears to document the longer timescale pH changes in the Pacific that cannot be archived from the marginal seas or expressed in reanalysis studies and model simulations.”

Line 298 and 300 Delete. I think this is a given. Data were generated a world-leading boron isotope lab. One would not expect sample wash out to be a considerable influence and the reader would not expect the authors to be interpreting trends that are within analytical error.

We have deleted the above sentences as suggested by this reviewer.

Line 375 replace “due to” with “caused by” Line 393 What does DPU stand for?

Original Line 375 words “due to” has been replaced with “caused by” as suggested by this Reviewer.

Original Line 393, the reference to the name DPU in the manuscript has now been removed. DPU is the internal name we gave for this coral colony based on the genus of *Diploastrea* (DP) and the location Uitoé (U).

Line 457 Unless this in-house standard has been measured elsewhere and has certified values the averages obtained here mean little to the reader and give no further indication of accuracy. Suggest leaving these results as precision only.

In this revised manuscript, the mean values of the in-house carbonate standard for $\delta^{18}\text{O}$ and $\delta^{13}\text{C}$ ratios have now been removed. The Reviewer is indeed correct that the values are not certified. The sentence stating this analytical precision can now be found on Line 478-480: “Long-term analytical precision based on repeated measurements of an in-house marble carbonate standard were $\pm 0.04\text{‰}$ ($\pm 1\sigma$ standard deviation, SD; $n = 116$) for $\delta^{18}\text{O}$ and $\pm 0.02\text{‰}$ ($\pm 1\sigma$ SD; $n = 116$) for $\delta^{13}\text{C}$ verified against NBS 19.”

Line 479 “in-house” repeated

This in-house standard for boron isotope analysis has not been certified in other analytical laboratories. This mention has now been removed from the revised manuscript and is only referred to in terms of its precision.

Line 501 Too many details in this section. Uptake rate of the particular neb used for example is of no use to a reader looking to replicate these results in their lab. I presume most of these details (e.g. cones) are given in the established methods of ref 57 and can therefore be cut here.

We agree that there is a lot of information contained in the Methods section of this manuscript. Based on the suggestion of the reviewer, we have removed particular sentences on the specific analytical procedure of the MC-ICP-MS.

Line 517 Was carry over a concern. How big were the blanks compared to samples? Was ammonia or HF used to help wash out?

We are absolutely certain that there was no carry-over (memory effect) from sample to sample during the analysis or from the extraction chemistry that was completed in batches. This procedure of randomizing the sample analysis was a precautionary measure, perhaps overcautious and unnecessary, but it was the easiest and most thorough way to completely avoid and eliminate any possible bias. In our bracketing “standard – sample” protocol, the rinse solution was composed of 0.1N HNO_3 and 0.04N HF and was used in between every measurement, this is stated in the Methods on original line 506. For extended information, we have revised the Methods on line

523 to state: “typical blank contribution is $< \sim 0.5\%$.”

Line 521 If JCt values are not quoted here they perhaps no need to mention them above. JCp is of course the more relevant standard here anyway.

The reference to the powder standard JCt-1 has now been removed and edited from the manuscript because the values were not reported.

Line 551 I fail to see the importance of using this fractionation factor calculation from the 1970s? Are the authors suggesting that the resultant low internal pH estimates are feasible? i.e. no pH upregulation at all.

Following this suggestion, we have modified the sentences in the Methods section. For clarification, the fractionation factor proposed by [Kakihana *et al.*, 1977] is now removed as the Reviewer is correct in that the isotopic fractionation factor proposed by [Klochko *et al.*, 2006] is probably the closest to that expected and also because the pH values derived from the use of this isotopic fractionation factor seems to be the most consistent with observations. The revised text can now be found on lines 565-570: “For this reconstruction, we applied the commonly used theoretically calculated fractionation factor ($\alpha_{[B3-B4]}$) of 1.0272 (ref. ³⁵) to facilitate comparison to previously published studies of $\delta^{11}\text{B}$ -pH reconstructions ¹²⁻¹⁷ and is also consistent with observations. The estimated seawater pH (pH_{sw}) value was then converted from the above calculated pH_c using the following equation based on *Porites* spp. from ref. ³⁹, $\text{pH}_{\text{sw}} = (\text{pH}_c - 5.954)/0.32$ (ref. ³⁹)(Fig. 2).”

Line 558 state here that this is the *Porites cylindrica* calibration from McCulloch Fig2 Could Figs 2 and 3 not be combined? I feel that Fig 3C adds little. Couldn't this smoother just be added to Fig 3B

The original Fig. 3 showing the conversion from coral $\delta^{11}\text{B}$ to pH in the (1) coral calcification space and then to (2) seawater pH has now been simplified and merged into the new Fig 2. The three stable isotope results are shown as before but with the addition of a reconstructed seawater pH y-axis. The original Fig. 3C was not a simple

smoothed record but is instead the calculated trend from our spectral analysis. The trends of each individual proxy record are now overlaid within each panel (dashed lines).

To clarify our method of pH conversion from coral $\delta^{11}\text{B}$, the Methods section has been edited to emphasize the *Porites* spp. calibration from [McCulloch et al., 2012a], which is the most similar to our coral (non-branching coral). This modified sentence can now be found on lines 568-570: “The estimated seawater pH (pH_{sw}) value was then converted from the above calculated pH_{c} using the following equation based on *Porites* spp. from ref. ³⁹, $\text{pH}_{\text{sw}} = (\text{pH}_{\text{c}} - 5.954)/0.32$ (ref. ³⁹)(Fig. 2)”

Fig 4 Remove word “record” from line labels. Make sure law dome label is closer to the grey line; at the moment it is not clear which line it corresponds to. Change axis label to "Atmospheric d13C"

The original Figure has been edited and revised following the suggestions of the Reviewer.

Fig 5 Circle markers on the plots are misleading. Are they in place of a legend or do they bear some relation to the x and y axes on their respective plots? I presume not. Perhaps these should be moved next to the d11B labels

Our original intent was not to overload the figure with legends or the names of each citation. Instead, we have chosen to use the coloured symbols to distinguish the individual dataset with matching locations inside the map that is at the top of the figure. To avoid further confusion for the reader, the marker dots are now moved to the y-axes next to the $\delta^{11}\text{B}$ labels.

Fig 7 caption states trend removed similar to fig 6. Should this be fig3?

This error has now been corrected and to be more precise for the reader, we have now clarified that it is not only “similar to Fig. 2” (new Fig. 2), it is in fact the “trend shown in Fig. 2”.

Fig S6 line 89 typo in the word “year”

This typo has now been corrected.

We thank this Reviewer for listing the positives of our work at the top of this review and noting the possible meaningful contribution to the field of coral-based geochemistry and ocean acidification research, as well as the thoughtful comments and recommendations that improved our manuscript.

References for Reply to Reviewer Comments:

- Cantin, N. E., and J. M. Lough (2014), Surviving Coral Bleaching Events: *Porites* Growth Anomalies on the Great Barrier Reef, *PLoS One*, 9(2), e88720.
- Comeau, S., E. Tambutté, R. C. Carpenter, P. J. Edmunds, N. R. Evensen, D. Allemand, C. Ferrier-Pagès, S. Tambutté, and A. A. Venn (2017), Coral calcifying fluid pH is modulated by seawater carbonate chemistry not solely seawater pH, *Proc. R. Soc. B Biol. Sci.*, 284(1847), doi:10.1098/rspb.2016.1669.
- Dishon, G., J. Fisch, I. Horn, K. Kaczmarek, J. Bijma, D. F. Gruber, O. Nir, Y. Popovich, and D. Tchernov (2015), A novel paleo-bleaching proxy using boron isotopes and high-resolution laser ablation to reconstruct coral bleaching events, *Biogeosciences*, 12(19), 5677–5687, doi:10.5194/bg-12-5677-2015.
- Gergis, J. L., and A. M. Fowler (2009), A history of ENSO events since A.D. 1525: implications for future climate change, *Clim. Change*, 92(3–4), 343–387, doi:10.1007/s10584-008-9476-z.
- Hathorne, E. C., T. Felis, A. Suzuki, H. Kawahata, and G. Cabioch (2013), Lithium in the aragonite skeletons of massive *Porites* corals: A new tool to reconstruct tropical sea surface temperatures, *Paleoceanography*, 28(1), 143–152, doi:10.1029/2012PA002311.
- Henley, B. J., J. Gergis, D. J. Karoly, S. Power, J. Kennedy, and C. K. Folland (2015), A Tripole Index for the Interdecadal Pacific Oscillation, *Clim. Dyn.*, 45(11–12), 3077–3090, doi:10.1007/s00382-015-2525-1.
- Huang, B., V. F. Banzon, E. Freeman, J. Lawrimore, W. Liu, T. C. Peterson, T. M. Smith, P. W. Thorne, S. D. Woodruff, and H.-M. Zhang (2014), Extended Reconstructed Sea Surface Temperature Version 4 (ERSST.v4). Part I: Upgrades and Intercomparisons, *J. Clim.*, 28(3), 911–930, doi:10.1175/JCLI-D-14-00006.1.
- Kakihana, H., M. Kotaka, S. Satoh, M. Nomura, and M. Okamoto (1977), Fundamental studies on the ion-exchange separation of Boron isotopes, *Bull. Chem. Soc. Jpn.*, 50(1), 158–163, doi:10.1246/bcsj.50.158.
- Klochko, K., A. J. Kaufman, W. Yao, R. H. Byrne, and J. A. Tossell (2006), Experimental measurement of boron isotope fractionation in seawater, *Earth Planet. Sci. Lett.*, 248(1–2), 276–285, doi:10.1016/j.epsl.2006.05.034.

- McCulloch, M., J. Falter, J. Trotter, and P. Montagna (2012a), Coral resilience to ocean acidification and global warming through pH up-regulation, *Nat. Clim. Chang.*, 2(8), 623–627, doi:10.1038/nclimate1473.
- McCulloch, M. et al. (2012b), Resilience of cold-water scleractinian corals to ocean acidification: Boron isotopic systematics of pH and saturation state up-regulation, *Geochim. Cosmochim. Acta*, 87, 21–34, doi:10.1016/j.gca.2012.03.027.
- McCulloch, M. T., J. P. D’Olivo, J. Falter, M. Holcomb, and J. A. Trotter (2017), Coral calcification in a changing World and the interactive dynamics of pH and DIC upregulation, *Nat. Commun.*, 8(May), 1–8, doi:10.1038/ncomms15686.
- Montagna, P. et al. (2014), Li/Mg systematics in scleractinian corals: Calibration of the thermometer, *Geochim. Cosmochim. Acta*, 132, 288–310, doi:10.1016/j.gca.2014.02.005.
- Quinn, T. M., T. J. Crowley, F. W. Taylor, C. Henin, P. Joannot, and Y. Join (1998), A multicentury stable isotope record from a New Caledonia coral: Interannual and decadal sea surface temperature variability in the southwest Pacific since 1657 A.D., *Paleoceanography*, 13(4), 412–426, doi:10.1029/98PA00401.
- Schoepf, V., M. T. McCulloch, M. E. Warner, S. J. Levas, Y. Matsui, M. D. Aschaffenburg, and A. G. Grottoli (2014), Short-term coral bleaching is not recorded by skeletal boron isotopes., *PLoS One*, 9(11), e112011, doi:10.1371/journal.pone.0112011.
- Tierney, J. E., N. J. Abram, K. J. Anchukaitis, M. N. Evans, C. Giry, K. H. Kilbourne, C. P. Saenger, H. C. Wu, and J. Zinke (2015), Tropical sea surface temperatures for the past four centuries reconstructed from coral archives, *Paleoceanography*, 30(3), 226–252, doi:10.1002/2014PA002717.
- Trotter, J., P. Montagna, M. McCulloch, S. Silenzi, S. Reynaud, G. Mortimer, S. Martin, C. Ferrier-Pagès, J.-P. Gattuso, and R. Rodolfo-Metalpa (2011), Quantifying the pH “vital effect” in the temperate zooxanthellate coral *Cladocora caespitosa*: Validation of the boron seawater pH proxy, *Earth Planet. Sci. Lett.*, 303(3–4), 163–173, doi:10.1016/j.epsl.2011.01.030.
- Wei, G., M. T. McCulloch, G. Mortimer, W. Deng, and L. Xie (2009), Evidence for ocean acidification in the Great Barrier Reef of Australia, *Geochim. Cosmochim. Acta*, 73(8), 2332–2346, doi:10.1016/j.gca.2009.02.009.
- Wei, G., Z. Wang, T. Ke, Y. Liu, W. Deng, X. Chen, J. Xu, T. Zeng, and L. Xie (2015), Decadal variability in seawater pH in the West Pacific : Evidence from coral $\delta^{11}\text{B}$ records, *J. Geophys. Res. Ocean.*, 120(11), 7166–7181, doi:10.1002/2015JC011066.
- Zeebe, R. E., and D. A. Wolf-Gladrow (2001), *CO₂ in Seawater: Equilibrium, Kinetics, Isotopes.*, Elsevier, Amsterdam.

REVIEWERS' COMMENTS:

Reviewer #1 (Remarks to the Author):

The authors have addressed my concerns and added figures and text to support their conclusions. I am satisfied with the changes and recommend publication as is.

Reviewer #2 (Remarks to the Author):

I feel that that the authors have done a satisfactory job in addressing the points raised during review and the manuscript is much improved.

Specifically, concerns that I raised about:

- 1) the choice of d11B-pH calibration,
 - 2) potential disturbance in the core coincident with the largest drop in d11B,
 - 3) discussion of pH increases
- have all been addressed.

I reiterate my previous summation of the paper that this is an important contribution that will be of interest to the wide readership of nature comms. I would now recommend the manuscript for publication.

I congratulate the authors on this piece of well executed research.

Joe Stewart

RESPONSE TO REVIEWER COMMENTS

REVIEWERS' COMMENTS:

Reviewer #1 (Remarks to the Author):

The authors have addressed my concerns and added figures and text to support their conclusions. I am satisfied with the changes and recommend publication as is.

Thank you very much for the helpful comments and suggestions in the revision.

Reviewer #2 (Remarks to the Author):

I feel that that the authors have done a satisfactory job in addressing the points raised during review and the manuscript is much improved.

Specifically, concerns that I raised about:

- 1) the choice of d11B-pH calibration,
 - 2) potential disturbance in the core coincident with the largest drop in d11B,
 - 3) discussion of pH increases
- have all been addressed.

I reiterate my previous summation of the paper that this is an important contribution that will be of interest to the wide readership of nature comms. I would now recommend the manuscript for publication.

I congratulate the authors on this piece of well executed research.

Joe Stewart

We thank Dr. Stewart for the kind words and the constructive comments and suggestions that significantly helped to improve this manuscript.